# ENHANCED DACER ALGORITHM WITH HIGH DIFFUSION EFFICIENCY

## ABSTRACT

Due to their expressive capacity, diffusion models have shown great promise in offline RL and imitation learning. Diffusion Actor-Critic with Entropy Regulator (DACER) extended this capability to online RL by using the reverse diffusion process as a policy approximator, achieving state-of-the-art performance. However, it still suffers from a core trade-off: more diffusion steps ensure high performance but reduce efficiency, while fewer steps degrade performance. This remains a major bottleneck for deploying diffusion policies in real-time online RL. To mitigate this, we propose DACERv2, which leverages a Q-gradient field objective with respect to action as an auxiliary optimization target to guide the denoising process at each diffusion step, thereby introducing intermediate supervisory signals that enhance the efficiency of single-step diffusion. Additionally, we observe that the independence of the Q-gradient field from the diffusion time step is inconsistent with the characteristics of the diffusion process. To address this issue, a temporal weighting mechanism is introduced, allowing the model to effectively eliminate large-scale noise during the early stages and refine its outputs in the later stages. Experimental results on OpenAI Gym benchmarks and multimodal tasks demonstrate that, compared with classical and diffusion-based online RL algorithms, DACERv2 achieves higher performance in most complex control environments with only **five diffusion steps** and shows greater multimodality.

## 1 INTRODUCTION

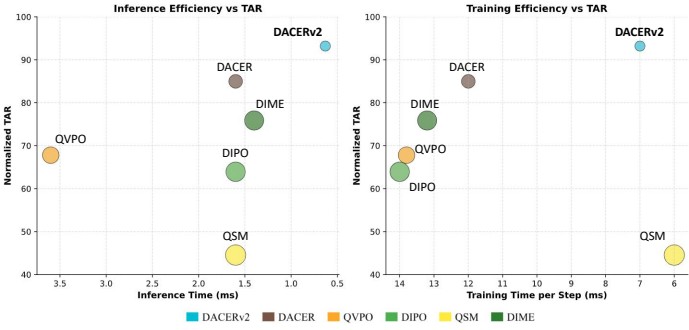

Figure 1: **Efficiency and Performance.** The horizontal axis represents the training or inference time (increasing from right to left), while the vertical axis shows the normalized Total Average Return (TAR). The training time is the per-step computational cost on OpenAI Gym tasks, excluding the time spent on environment interaction. The inference time is measured as the latency required for the policy network to output an action given a single state as input. DACERv2 achieve outstanding performance.

Energy-based models are well-suited as agent policy functions due to their powerful representational capabilities. Learning a policy to approximate the corresponding energy-based target distribution allows for modeling complex and multimodal action patterns without relying on restrictive parametric assumptions, especially in continuous action spaces. This enhanced expressiveness can significantly improve exploration by enabling the agent to discover and leverage diverse behavioral strategies.

However, effectively approximating such an expressive soft policy presents notable challenges. A key difficulty lies in how to efficiently and accurately sample from the target distribution. While algorithms such as Soft Actor-Critic (SAC) (Haarnoja et al., 2018) and Distributional Soft Actor-Critic (DSAC) (Duan et al., 2021; 2025) aim to approximate the soft-target distribution, they typically represent the policy as a simple Gaussian, enabling analytical entropy computation. This choice is computationally efficient but falls short in modeling complex and multimodal behavior. Meanwhile, due to their strong representational capacity, diffusion models have emerged as a promising policy class for continuous control, commonly referred to as diffusion policies (Ren et al., 2024; Li et al., 2024; Lu et al., 2025b).

Existing methods for training diffusion policies can be broadly categorized into two groups: score-matching and end-to-end policy gradient approaches. In the first group, QVPO (Ding et al., 2024) proposes using Q-weighted imitation learning samples to improve policy learning. QSM (Psenka et al., 2023) directly aligns the score functions with the gradients of the learned Q-functions and uses Langevin dynamics for sampling. DIPO (Yang et al., 2023a) updates the replay buffer using action gradients and improves the performance of the policy through a diffusion loss. In the second group, DACER (Wang et al., 2024) directly backward the gradient through the reverse diffusion process and proposes a Gaussian mixture model (GMM) entropy regulator to balance exploration and exploitation. DIME (Celik et al., 2025) derives an approximate maximum-entropy lower bound, directly integrating the maximum-entropy RL framework with the diffusion policy. However, diffusion policies typically require a large number of diffusion steps to maintain strong performance, which results in low inference efficiency. Although acceleration techniques such as DPM-Solver (Lu et al., 2022) can reduce the number of diffusion steps, this often comes at the cost of performance degradation. As a result, previous methods struggle to escape the dilemma between performance and time-efficiency.

To tackle this issue, we present DACERv2, a highly efficient diffusion-based RL algorithm that achieves comparable or superior performance with only a few diffusion steps, as shown in Fig. 1. The key contributions of this paper are the following: 1) We propose a Q-gradient field objective as an extra intermediate supervisory signal to enhance the efficiency of single-step diffusion. 2) Since the Q-gradient field is independent of the diffusion time, we propose a temporal weighting mechanism that takes the current diffusion time step as input. This mechanism aligns with the requirements of the diffusion denoising process: higher denoising amplitudes during early stages and lower denoising amplitudes for precise control in later stages. 3) We evaluate the performance of our method on the OpenAI Gym benchmark (Brockman, 2016). Compared with both diffusion-based and classical algorithms like DACER (Wang et al., 2024), QVPO (Ding et al., 2024), DIME (Celik et al., 2025), QSM (Psenka et al., 2023), DIPO (Yang et al., 2023a), DSAC (Duan et al., 2025), PPO (Schulman et al., 2017), and SAC (Haarnoja et al., 2018), our approach achieved state-of-the-art (SOTA) performance in most complex control tasks. 4) We evaluate the training and inference times of all diffusion-based algorithms under identical hardware configurations using the PyTorch framework. While achieving stronger overall performance, our method reduces training time by **47.0%** and **41.7%**, and inference time by **55.0%** and **60.6%**, compared with DIME and DACER, respectively.

## 2 PRELIMINARIES

### 2.1 REINFORCEMENT LEARNING WITH SOFT POLICY

RL problems are commonly modeled as Markov Decision Processes (MDPs) (Sutton & Barto, 2018; Li, 2023). An infinite-horizon MDP is defined by a tuple $(\mathcal{S}, \mathcal{A}, P, r, \gamma)$, where $\mathcal{S}$ is the state space and $\mathcal{A}$ is the action space, both assumed bounded and potentially continuous. $P : \mathcal{S} \times \mathcal{A} \mapsto \Delta(\mathcal{S})$ denotes the transition dynamics, specifying the probability distribution $P(\cdot \mid s_t, a_t)$ over next states, with $\Delta(\mathcal{S})$ representing the set of distributions over $\mathcal{S}$. $r : \mathcal{S} \times \mathcal{A} \mapsto \mathbb{R}$ is the reward function, and $\gamma \in [0, 1)$ is the discount factor. The behavior of agent is characterized by a policy $\pi : \mathcal{S} \mapsto A$, which defines the process of selecting action $a$ given the state $s$. To evaluate the value of taking an action $a$ in a given state $s$ under policy $\pi$, the action-value function $Q^\pi(s, a)$ is introduced, which represents the expected cumulative discounted reward, defined as follows:

$$Q^\pi(s, a) = \mathbb{E}_\pi \Big[ \sum_{i=0}^{\infty} \gamma^i r(s_i, a_i) \mid s_0 = s, a_0 = a \Big]. \tag{1}$$

A key challenge in online RL is the trade-off between exploration, gathering information for future gains, and exploitation, maximizing returns based on current knowledge. One compelling strategy involves learning a policy that aims to approximate a soft policy (Haarnoja et al., 2017; 2018; Ma et al., 2025; Messaoud et al., 2024). Such target soft policies are typically formulated as a Boltzmann distribution, where the desired policy distribution is proportional to the exponentiated state-action value function:

$$\pi_{\text{soft}}(a|s) \propto \exp\left(\frac{1}{\alpha}Q(s,a)\right). \tag{2}$$

The target of soft policy is to minimize the per-state KL divergence $D_{\text{KL}}\left(\pi(\cdot|s) \,\|\, \frac{\exp(Q(s,\cdot)/\alpha)}{Z(s)}\right)$, where $Z(s)$ is the normalization coefficient. This KL-divergence minimization problem is equivalent to maximizing a final objective function that balances value maximization and entropy regularization:

$$J(\pi) = \mathbb{E}_{(s,a)\sim\pi}\left[Q(s,a)\right] + \alpha \cdot \mathcal{H}(\pi(\cdot|s)). \tag{3}$$

Diffusion policies are able to model complex policy distributions, but their entropy is analytically intractable. Fortunately, in methods like DACER (Wang et al., 2024), maximizing the $Q$-value objective under specific entropy regularization likewise produces a Boltzmann policy. See Theorem 1 for further theoretical details.

## 2.2 DIFFUSION MODELS AS EXPRESSIVE POLICY

Diffusion models (Ho et al., 2020; Song et al., 2020b; Wang et al., 2024) conceptualize data generation as a stochastic process where data samples are iteratively reconstructed via a parameterized reverse-time stochastic differential equation (SDE). Although both forward and reverse diffusion processes are theoretically integral to these models, recent work (Chen et al., 2024) highlights that their expressive power primarily stems from the reverse-time denoising dynamics, rather than the forward-time noising process. Accordingly, our analysis and modeling efforts concentrate on the reverse diffusion process.

Formally, the continuous reverse-time SDE that governs this process is defined as follows:

$$d\boldsymbol{x} = \left[f(\boldsymbol{x},t) - g(t)^2 \nabla_{\boldsymbol{x}} \log p_t(\boldsymbol{x})\right] dt + g(t)\, d\omega(t), \tag{4}$$

where $f(\boldsymbol{x},t)$ represents the drift term, $g(t)$ denotes the time-dependent diffusion coefficient, $\nabla_{\boldsymbol{x}} \log p_t(\boldsymbol{x})$ is the score function, and $d\omega(t)$ is the standard Wiener process. The term $\nabla_{\boldsymbol{x}} \log p_t(\boldsymbol{x})$, also known as the score function, plays a central role in guiding the reverse diffusion dynamics. It is important to note that this equation represents the general form of the reverse-time SDE; the specific construction of terms such as $f(\boldsymbol{x},t)$ and $g(t)$ can vary across different diffusion model algorithms.

Therefore, the reverse-time SDE of diffusion policy can be expressed as:

$$d\boldsymbol{a}_t = \left[f(\boldsymbol{a}_t,t) - g(t)^2 S_\theta(\boldsymbol{s},\boldsymbol{a}_t,t)\right] dt + g(t)\, d\omega(t), \tag{5}$$

where $S_\theta(\boldsymbol{s},\boldsymbol{a}_t,t)$ is a neural network designed to approximate the gradient $\nabla_{\boldsymbol{a}_t} \log p_t(\boldsymbol{a}_t|\boldsymbol{s})$. Actions can be sampled from the diffusion policy $\pi_\theta(\boldsymbol{a}_0|\boldsymbol{s})$ by solving the following integral:

$$\boldsymbol{a}_0 = \boldsymbol{a}_T + \int_0^T \left[f(\boldsymbol{a}_\tau,\tau) - g(\tau)^2\, S_\theta(\boldsymbol{s},\mathbf{a}_\tau,\tau)\right] d\tau + \int_0^T g(\tau)\, d\omega(\tau), \tag{6}$$

where $\boldsymbol{a}_T$ follows a standard normal distribution $\mathcal{N}(0,\boldsymbol{I})$.

## 2.3 LANGEVIN DYNAMICS

Langevin dynamics represents a powerful computational framework for simulating particle motion under the joint influence of deterministic forces and stochastic fluctuations. When coupled with stochastic gradient descent, this approach gives rise to stochastic gradient Langevin dynamics (SGLD) (Welling & Teh, 2011) - an efficient sampling algorithm that leverages log-probability gradients $\nabla_{\boldsymbol{x}} \log p(\boldsymbol{x})$ to draw samples from probability distributions $p(x)$ through an iterative Markov chain process:

$$\boldsymbol{x}_{t-1} = \boldsymbol{x}_t + \frac{\delta_t}{2}\nabla_{\boldsymbol{x}_t} \log p(\boldsymbol{x}_t) + \sqrt{\delta_t}\boldsymbol{\epsilon}, \tag{7}$$

where $\boldsymbol{\epsilon} \sim \mathcal{N}(\mathbf{0},\boldsymbol{I})$, $\delta_t$ is the step size. When $t$ range from infinity to one, $\delta_t \to 0$, $x_0$ equals to the true probability density $p(x)$.

## 3 METHOD

In this section, we explain how our method approximates the target policy distribution with fewer diffusion steps. First, we show that $\nabla_{\boldsymbol{a}_t} Q(\boldsymbol{s}, \boldsymbol{a}_t)$, derived from Langevin dynamics, can be incorporated into the unified SDE-based framework for action generation, thereby improving the efficiency of single-step diffusion. However, when this extra objective function is incorporated, the diffusion policy only exhibits suboptimal performance. This limitation arises because $\nabla_{\boldsymbol{a}_t} Q(\boldsymbol{s}, \boldsymbol{a}_t)$ remains independent of the diffusion step, whereas the score function is not. Therefore, we introduce a time-weighted mechanism to better align with the requirements of the diffusion denoising process. Lastly, we propose a practical algorithm for optimizing diffusion models.

### 3.1 Q-GRADIENT FIELD GUIDED DENOISING

Using the only reverse process, the objective function of DACER is to maximize the Q-value, representing an end-to-end optimization approach without direct supervision in the intermediate diffusion steps. However, within this optimization scheme, the guidance signals at intermediate steps are implicit, as they are derived solely from the final Q-value through back-propagation, which in turn necessitates more diffusion steps to produce higher-quality control actions. To address this issue, we propose the Q-gradient field function as an extra training loss to enhance the efficiency of single-step diffusion. At the end of Section 2.1, we explain why, when the global policy entropy is fixed, the optimal policy for maximizing the Q-value theoretically follows a Boltzmann distribution. Importantly, this conclusion holds for policy families of arbitrary forms and naturally suits the SDE-based policy families.

From another perspective, Langevin dynamics can be regarded as a special form of an SDE-based policy, providing an efficient method for sampling actions from Boltzmann distributions (Hinton, 2002):

$$\pi(\boldsymbol{a}|\boldsymbol{s}) = \frac{e^{\frac{1}{\alpha} Q(\boldsymbol{s}, \boldsymbol{a})}}{Z(\boldsymbol{s})}, \tag{8}$$

where $\alpha > 0$ is the temperature coefficient, $Q(\boldsymbol{s}, \boldsymbol{a})$ is the state action value function and $Z(\boldsymbol{s})$ is the partition function that normalizes the distribution. The formula derived by taking the partial derivative of both sides of Eq. (8) with respect to $\boldsymbol{a}$ can be expressed as

$$\nabla_{\boldsymbol{a}} \log \pi(\boldsymbol{a}|\boldsymbol{s}) = \frac{1}{\alpha} \nabla_{\boldsymbol{a}} Q(\boldsymbol{s}, \boldsymbol{a}). \tag{9}$$

Substituting Eq. (9) into Eq. (7), we can obtain the sampling process for $\pi(\boldsymbol{a}|\boldsymbol{s})$:

$$\boldsymbol{a}_{t-1} = \boldsymbol{a}_t + \frac{\delta_t}{2\alpha} \nabla_{\boldsymbol{a}} Q(\boldsymbol{s}, \boldsymbol{a}) + \sqrt{\delta_t} \boldsymbol{\epsilon}. \tag{10}$$

In summary, Langevin dynamics can be regarded as a particular solution within the family of SDE-based policies. This connection motivates the use of $\nabla_{\boldsymbol{a}} Q(\boldsymbol{s}, \boldsymbol{a})$ as an extra learning objective to guide the training of SDE-based policies, thereby introducing additional supervision signals into the intermediate diffusion step. Consequently, the efficiency of single-step diffusion can be improved, enabling comparable or even superior performance to previous algorithms with fewer diffusion steps.

Moreover, in highly unstable environments that exhibit extreme sensitivity to minor action perturbations, the Q-gradient estimation can become volatile, potentially hindering the algorithm's convergence to the optimal policy (Ding et al., 2024; Ma et al., 2025). When the diffusion process is restricted to only a few steps, a policy trained solely on the Q-gradient often struggles to converge. For these reasons, we adopt it only as an auxiliary guidance in policy training.

### 3.2 TIME-WEIGHTED MECHANISM

In the previous subsection, we propose incorporating the gradient term $\nabla_{\boldsymbol{a}_t} Q(\boldsymbol{s}, \boldsymbol{a}_t)$ as an auxiliary objective when training the SDE-based policy. However, experimental results show that directly employing this objective yields suboptimal performance, as shown in Fig. 4(b). We attribute this to the Q-gradient being independent of the diffusion time step, whereas the score function is not. Such time invariance fails to satisfy the varying denoising requirements across the diffusion process.

Specifically, in the later stages of diffusion process, the denoising intensity should naturally decrease as the action distribution approaches the optimal policy.

To address this issue, we introduce a time-weighted mechanism that modulates the influence of Q-gradient guidance based on the diffusion time step, allowing for more precise control over the denoising process. Inspired by the design approach for the step size $\delta_t$ in Eq. (7), we can similarly design our time-weighted mechanism using the commonly employed exponential decay function (Welling & Teh, 2011; Teh et al., 2016):

$$w(t) = \exp(c \cdot t + d), \tag{11}$$

where $t$ denotes the current diffusion step. The hyperparameters $c$ and $d$ are chosen inspired by the variance-preserving beta schedule in DDPM (Ho et al., 2020) and depend only on the number of diffusion steps. The specific setting is presented in Appendix D.

Furthermore, to improve the stability of the training process, we normalize $\nabla_{\boldsymbol{a}_t} Q(\boldsymbol{s}, \boldsymbol{a}_t)$ by its norm:

$$\nabla_{\boldsymbol{a}_t} Q_{\text{norm}}(\boldsymbol{s}, \boldsymbol{a}_t) = \frac{\nabla_{\boldsymbol{a}_t} Q(\boldsymbol{s}, \boldsymbol{a}_t)}{||\nabla_{\boldsymbol{a}_t} Q(\boldsymbol{s}, \boldsymbol{a}_t)|| + \epsilon}, \tag{12}$$

where $\epsilon$ is a small constant to prevent division by zero.

Ultimately, we construct the Q-gradient field objective function to facilitate the training of the diffusion policy, where $\pi_\theta(\boldsymbol{a}_t|\boldsymbol{s})$ denotes the action generated using the diffusion policy as defined in Eq. (6):

$$\mathcal{L}_g(\theta) = \min_\theta \mathbb{E}_{\substack{\boldsymbol{s} \sim \mathcal{B} \\ t \sim \text{U}(1,T) \\ \boldsymbol{a}_t \sim \pi_\theta(\boldsymbol{a}_t|\boldsymbol{s})}} \left[ \|w(t)\nabla_{\boldsymbol{a}_t} Q_{\text{norm}}(\boldsymbol{s}, \boldsymbol{a}_t) - S_\theta(\boldsymbol{s}, \boldsymbol{a}_t, t)\|_2^2 \right], \tag{13}$$

where U means uniform distribution, $t$ is the current diffusion step, $\mathcal{B}$ represents the replay buffer, and $\theta$ is the network parameter of the diffusion policy. The subscript $g$ represents the objective function related to the Q-gradient.

### 3.3 DACERv2: A High Efficiency Diffusion RL Algorithm

To obtain a practical algorithm, we use a parameterized function approximation for the Q-function and the diffusion policy. In the critic component, we adopt the double Q-learning strategy (Fujimoto et al., 2018) to alleviate overestimation bias. Specifically, we maintain two independent Q-function estimators, denoted as $Q_{\phi_1}(\boldsymbol{s}, \boldsymbol{a})$ and $Q_{\phi_2}(\boldsymbol{s}, \boldsymbol{a})$, which are trained to approximate the true action-value function. To enhance training stability, we introduce two corresponding target networks, $Q_{\bar{\phi}_1}(\boldsymbol{s}, \boldsymbol{a})$ and $Q_{\bar{\phi}_2}(\boldsymbol{s}, \boldsymbol{a})$, which are updated softly from the main networks following the technique in (Van Hasselt et al., 2016).

The Q-networks are optimized by minimizing the Bellman error. For each network $Q_{\phi_i}(\boldsymbol{s}, \boldsymbol{a})$, the loss $J_Q(\phi_i)$ is defined as:

$$J_Q(\phi_i) = \mathbb{E}_{\substack{(\boldsymbol{s}, \boldsymbol{a}, r, \boldsymbol{s}') \sim \mathcal{B} \\ \boldsymbol{a}' \sim \pi_\theta(\boldsymbol{a}_0|\boldsymbol{s})}} \left[ \left( \left( r(\boldsymbol{s}, \boldsymbol{a}) + \gamma \min_{i=1,2} Q_{\bar{\phi}_i}(\boldsymbol{s}', \boldsymbol{a}') \right) - Q_{\phi_i}(\boldsymbol{s}, \boldsymbol{a}) \right)^2 \right], \tag{14}$$

where $\gamma$ is discount factor, the target is computed as the smaller of the two target Q-values, $Q_{\bar{\phi}_1}(\boldsymbol{s}', \boldsymbol{a}')$ and $Q_{\bar{\phi}_2}(\boldsymbol{s}', \boldsymbol{a}')$, to prevent over-optimistic estimates. Furthermore, we incorporate the distributional value estimation framework from DSAC (Duan et al., 2025) to further mitigate overestimation issues.

In the actor component, we follow the objective function of maximizing the Q value and combine it with the auxiliary learning objective based on the Q-gradient field proposed in this paper. The final policy-learning objective is a linear combination:

$$\pi = \arg\min_{\pi_\theta} \mathcal{L}_\pi(\theta) = \mathcal{L}_q(\theta) + \eta \cdot \mathcal{L}_g(\theta),$$
$$\text{s.t.} \quad \mathbb{E}_{s \sim p(s)}[H(\pi^*(\cdot|s))] = \mathcal{H}^{\text{target}}, \tag{15}$$

where $\eta$ is a hyperparameter, $\mathcal{L}_q(\theta) = \mathbb{E}_{\boldsymbol{s} \sim \mathcal{B}, \boldsymbol{a}_0 \sim \pi_\theta(\cdot|\boldsymbol{s})} \left[ -Q_\phi(\boldsymbol{s}, \boldsymbol{a}_0) \right]$, $p(s)$ is a distribution over states. $\pi^*$ is the Boltzmann-optimal policy under the global entropy $\mathcal{H}^{\text{target}}$. We adopt the entropy regularization method from the original DACER algorithm to control the global policy entropy.

## 4 EXPERIMENTAL RESULTS

Multimodality is a key metric for evaluating diffusion-based algorithms. Therefore, we first validate DACERv2 with respect to this metric in the "Multi-goal" environment (Haarnoja et al., 2017), as illustrated in Fig. 2. We then conducted experiments on eight tasks in OpenAI Gym MuJoCo (Brockman et al., 2016). These environments represent challenging learning tasks with action spaces of up to 17 dimensions and observation spaces of up to 376 dimensions. With these experimental results, we aim to answer three questions:

- Does DACERv2 demonstrate stronger multimodal capabilities?
- How does the inference and training efficiency of DACERv2 compare with existing diffusion-based RL methods?
- How does DACERv2 compare to previous popular online RL algorithms and existing diffusion-based online RL algorithms?

**Baselines.** The baselines encompass two categories of model-free reinforcement learning algorithms. The first category consists of diffusion-based RL methods, including a range of recent diffusion-policy online algorithms such as DACER (Wang et al., 2024), QVPO (Ding et al., 2024), DIME (Celik et al., 2025), DIPO (Yang et al., 2023b), and QSM (Psenka et al., 2023). The second category includes classic model-free online RL baselines, namely SAC (Haarnoja et al., 2018), PPO (Schulman et al., 2017), and DSAC (Duan et al., 2025). The experimental hyperparameters are provided in Appendix D. It is worth noting that the Critic network in DIME employs a two-layer MLP with a hidden dimension of 2048, consistent with their original paper, whereas the corresponding dimension for other algorithms is 256.

**Evaluation Setups.** We implemented our algorithm in PyTorch and evaluated it on eight MuJoCo tasks using the same metrics as DACER. Experiments were conducted on a system equipped with an AMD Ryzen Threadripper 3960X 24-core processor and an NVIDIA GeForce RTX 4090 GPU. In this paper, the total training step size for all experiments was set at 1.5 million, with the results of all experiments averaged over 5 random seeds. For classic model-free baselines, we cited DACER-reported results, while all diffusion-based methods were re-evaluated. Furthermore, the training curves presented in Fig. 3 demonstrate the stability of the training process.

### 4.1 MULTIMODAL EXPERIMENTS

We evaluate the trained policy in the "Multi-goal" environment by initializing the agent at the origin and sampling 100 trajectories. We conducte three sets of experiments with configurations ranging from 4 to 6 symmetrically arranged goal points. As illustrated in Fig. 2, the original DACER algorithm fails to maintain uniform coverage as the number of target points increases; when six targets are specified, the algorithm reaches only five target goals. In contrast, our method consistently reaches all target locations with approximately uniform coverage. These experimental results underscore that our method achieves superior exploratory capability, enabling it to more effectively capture diverse, mode-separated policies in multimodal environments.

### 4.2 EFFICIENCY ANALYSIS

We first define the training time as the per-step computational cost on MuJoCo tasks, excluding the time spent on environment interaction. The inference time is measured as the latency required for the policy network to output an action given a single state as input. As illustrated in Table 1, the inference times of DACER, QVPO, DIME, DIPO, and QSM are **2.54×, 5.71×, 2.22×, 2.54×**, and **2.54×** longer than our method, respectively. For training time, their costs are **1.71×, 1.97×, 1.89×, 2.00×**, and **0.86×** relative to our method. Since our method achieves markedly superior performance compared to QSM, its slight disadvantage in training time is negligible in practice.

These results can be attributed to the use of a Q-gradient field objective as an auxiliary intermediate supervisory signal, which enhances the efficiency of single-step diffusion and enables our algorithm to achieve competitive performance with only five diffusion steps.

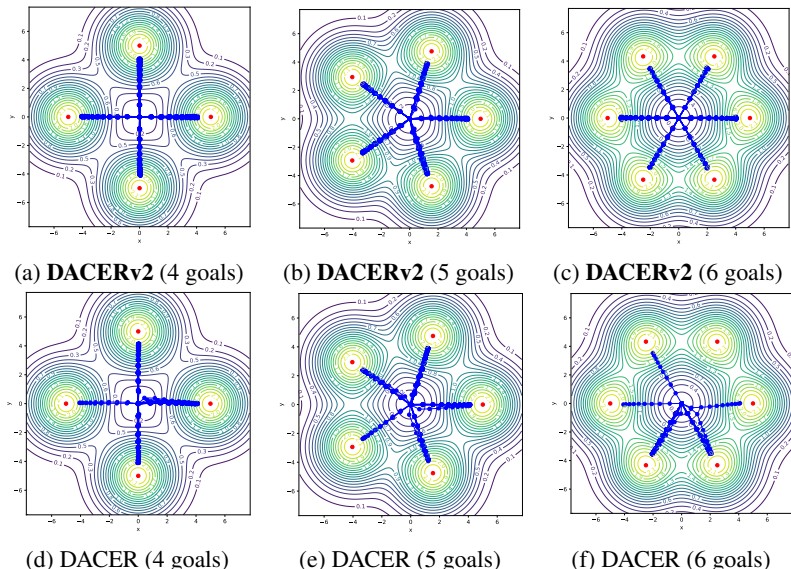

| (a) **DACERv2** (4 goals) | (b) **DACERv2** (5 goals) | (c) **DACERv2** (6 goals) |
|:-:|:-:|:-:|
| (d) DACER (4 goals) | (e) DACER (5 goals) | (f) DACER (6 goals) |

Figure 2: **Multi-goal Task.** Trajectories generated by policies learned using our method (top row) and original DACER (bottom row) are shown, with the $x$-axis and $y$-axis representing 2D positions (states). The agent is initialized at the origin, and the goals are marked as red dots. The level curves indicate the reward, and reaching within 1 of the endpoint signifies task completion. Results are shown for 4, 5, and 6 goal configurations from left to right.

Table 1: Efficiency comparison of inference and training time. All values are normalized relative to **DACERv2** (set as $1.00\times$). Absolute times are also reported. Lower is better.

| Algorithms | Inference Time | | Training Time | |
|---|---|---|---|---|
| | **Normalized** | **Absolute (ms)** | **Normalized** | **Absolute (ms)** |
| **DACERv2 (Ours)** | $1.00\times$ | 0.63 | $1.00\times$ | 7.0 |
| DACER | $2.54\times$ | 1.60 | $1.71\times$ | 12.0 |
| QVPO | $5.71\times$ | 3.60 | $1.97\times$ | 13.8 |
| DIME | $2.22\times$ | 1.40 | $1.89\times$ | 13.2 |
| DIPO | $2.54\times$ | 1.60 | $2.00\times$ | 14.0 |
| QSM | $2.54\times$ | 1.60 | $0.86\times$ | 6.0 |

## 4.3 EXPERIMENTAL RESULTS

All the training curves are shown in Fig. 3 and the detailed results are listed in Table 2. Our method, DACERv2, achieves superior Total Average Return (TAR) in most complex OpenAI Gym control tasks. Despite the challenges posed by high-dimensional state and action spaces and complex dynamics, our method exhibits remarkable stability and efficiency, highlighting its robustness and adaptability.

Specifically, across challenging environments including Humanoid, Ant, HalfCheetah, Humanoid-Standup, and Walker2d, our method achieves improvements of **33.1%, 42.7%, 9.8%, 5.9%**, and **29.2%** over SAC, respectively. When compared against the best-performing diffusion-based online RL baseline in each environment, it achieves higher scores in Ant, HalfCheetah, HumanoidStandup, and Walker2d, with respective gains of **4.3%, 4.0%, 5.6%**, and **10.3%**, while underperforming DIME on Humanoid. Additionally, we normalize the returns in each task by dividing them by the highest reward across all algorithms, then average across tasks and rescale to the range of 0–100 for visualization. Under this metric, our method achieves an average score **9.7%** higher than the second-best algorithm on the OpenAI Gym benchmark.

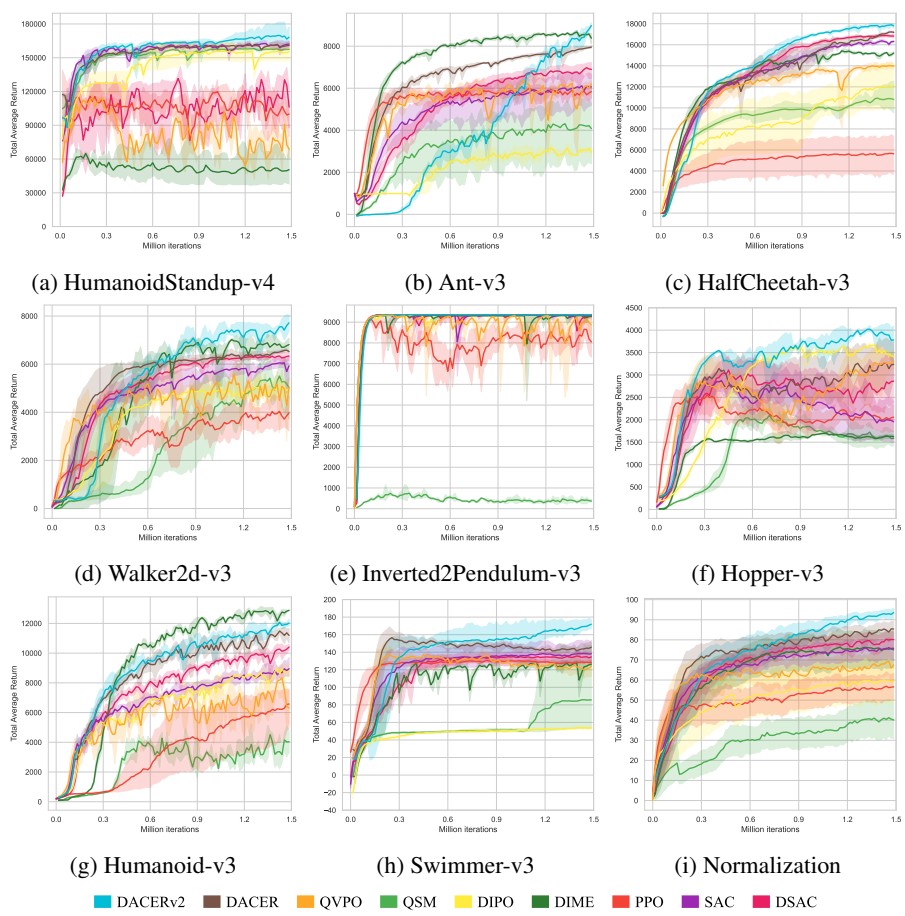

Figure 3: **Training curves on benchmarks.** The solid lines represent the mean, while the shaded regions indicate the 95% confidence interval over five runs. For PPO, iterations are defined by the number of network updates.

Table 2: **Total Average Return (TAR).** Performance on eight tasks of OpenAI Gym MuJoCo benchmark. Mean ± Std. over 5 seeds. **Bold** = best; higher is better. The average score has been normalized to the range of 0-100.

| Algorithm | HumanoidStandup | Ant | Humanoid | Walker2d | Inverted2Pendulum | Hopper | HalfCheetah | Swimmer | Average score |
|---|---|---|---|---|---|---|---|---|---|
| PPO | $82807 \pm 8633$ | $6157 \pm 185$ | $6869 \pm 1563$ | $4832 \pm 638$ | $9357 \pm 2$ | $2647 \pm 481$ | $5789 \pm 2201$ | $130 \pm 2$ | $56.69 \pm 19.80$ |
| SAC | $161413 \pm 1643$ | $6427 \pm 804$ | $9335 \pm 696$ | $6201 \pm 263$ | $\mathbf{9360 \pm 0}$ | $2483 \pm 943$ | $16573 \pm 224$ | $140 \pm 14$ | $75.41 \pm 17.64$ |
| DSAC | $149576 \pm 1795$ | $7086 \pm 261$ | $10829 \pm 243$ | $6424 \pm 147$ | $\mathbf{9360 \pm 0}$ | $3660 \pm 533$ | $17025 \pm 157$ | $138 \pm 6$ | $80.18 \pm 12.07$ |
| QSM | $150692 \pm 1497$ | $4783 \pm 1235$ | $6072 \pm 691$ | $5685 \pm 437$ | $591 \pm 98$ | $2006 \pm 251$ | $11401 \pm 882$ | $46 \pm 1$ | $44.54 \pm 25.05$ |
| DIPO | $156870 \pm 8270$ | $3449 \pm 149$ | $9353 \pm 356$ | $5066 \pm 365$ | $9355 \pm 2$ | $3813 \pm 241$ | $12267 \pm 2180$ | $55 \pm 2$ | $63.93 \pm 23.00$ |
| DIME | $78303 \pm 3165$ | $8789 \pm 105$ | $\mathbf{13065 \pm 221}$ | $7261 \pm 299$ | $9356 \pm 2$ | $2016 \pm 179$ | $15816 \pm 292$ | $134 \pm 3$ | $75.87 \pm 22.34$ |
| DACER | $161928 \pm 3804$ | $8040 \pm 128$ | $11791 \pm 238$ | $6674 \pm 169$ | $9354 \pm 2$ | $4062 \pm 181$ | $17488 \pm 216$ | $150 \pm 4$ | $84.98 \pm 11.38$ |
| QVPO | $129865 \pm 8932$ | $6484 \pm 145$ | $9656 \pm 252$ | $6057 \pm 352$ | $9354 \pm 5$ | $4035 \pm 172$ | $14355 \pm 175$ | $130 \pm 10$ | $67.80 \pm 16.74$ |
| **DACERv2 (ours)** | $\mathbf{170956 \pm 8792}$ | $\mathbf{9169 \pm 129}$ | $12426 \pm 292$ | $\mathbf{8011 \pm 188}$ | $9359 \pm 1$ | $\mathbf{4202 \pm 191}$ | $\mathbf{18192 \pm 266}$ | $\mathbf{172 \pm 6}$ | $\mathbf{93.19 \pm 6.35}$ |

## 4.4 ABLATION STUDY

In this section, we conduct ablation study to investigate the impact of the following four aspects on the performance of the diffusion policy: 1) whether to use the Q-gradient field training objective function; 2) whether to use time-weighted mechanism; 3) different diffusion step size $T$; 4) the sensitivity to the hyperparameter $\eta$. The experiments are conducted in the Walker2d-v3 task. Ablation study on the effect of Q-value normalization is provided in Appendix F.

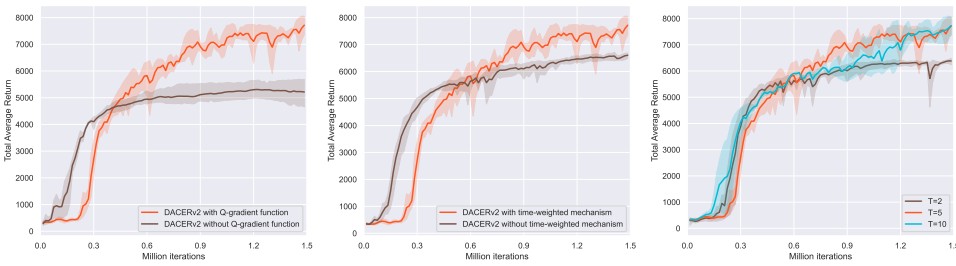

(a) Ablation for the Q-gradient field training objective function.

(b) Ablation for the time-weighted mechanism.

(c) Ablation for the different diffusion steps.

Figure 4: **Ablation experiment curves.** (a) The performance of DACERv2 with Q-gradient function on Walker2d-v3 is far better than without Q-gradient function. (b) Time-weighted mechanism can further improve the performance of our algorithm. (c) A diffusion step size of 5 provides a balance between efficiency and performance.

**Q-gradient field training objective function.** In this ablation study, we fixed the diffusion step size at 5 to examine the effect of incorporating the Q-gradient field loss function. As shown in Fig. 4(a), removing this objective caused a substantial drop in performance. This finding highlights the critical role of the Q-gradient field loss in guiding the diffusion denoising process and demonstrates its importance as a key component for enhancing overall performance.

**Time-weighted mechanism.** We conducted an experiment to demonstrate that using time-weighted mechanism can further improve performance. As shown in Fig. 4(b), directly using $\nabla_a Q(s, a)$ as the target value in the Q-gradient field training loss, instead of the $w(t)\nabla_a Q(s, a)$, results in performance degradation. This is because different timesteps require matching different magnitudes of noise prediction, which enhances both training stability and final performance.

**Diffusion steps.** We further investigated the performance of the diffusion policy under varying numbers of diffusion timesteps $T$. We plotted training curves for $T = 2, 5$, and $10$, as shown in Fig. 4(c). The experimental results suggest that increasing the number of diffusion steps does not necessarily improve performance, while using fewer steps tends to degrade performance.

**The sensitivity to the hyperparameter $\eta$.** To assess the sensitivity of $\eta$, we evaluated five settings $(0.1, 0.01, 0.001, 0.012, 0.008)$ on Humanoid-v3. As reported in Table 3, performance degraded markedly at $\eta = 0.1$ and $0.001$, but remained stable at $\eta = 0.012$ and $0.008$, indicating tenfold sensitivity. These results suggest that the algorithm is robust to moderate variations in $\eta$ and thus does not require extensive hyperparameter tuning.

Table 3: Performance comparison of DACERv2 with different $\eta$ values on Humanoid-v3.

| Algorithm | $\eta = 0.01$ | $\eta = 0.1$ | $\eta = 0.001$ | $\eta = 0.012$ | $\eta = 0.008$ |
|---|---|---|---|---|---|
| **DACERv2** | **12426 $\pm$ 292** | 11463 $\pm$ 304 | 11161 $\pm$ 287 | 12101 $\pm$ 325 | 12208 $\pm$ 249 |

## 5 CONCLUSION

In this paper, we address the critical challenge of balancing performance and time-efficiency in diffusion-based online RL. By introducing a Q-gradient field objective and a time-dependent weighting scheme, our method enables each denoising step to be guided by the Q-function with adaptive emphasis over time. This design allows the policy to achieve strong performance with only five diffusion steps, significantly improving both training and inference speed.

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

## A    THEORETICAL ANALYSIS

**Theorem 1.** *Let $\mathcal{S}$ denote the state space and $\mathcal{A}$ denote the continuous action space. Suppose $p(s)$ is a distribution over states, $\mathcal{H}_0^{global}$ denotes a specific entropy value. We define the policy space $\Pi_{\mathcal{H}_0^{global}}$ as the set of policy families $\{\pi^*(\cdot|s)\}_{s\in\mathcal{S}}$, where each $\pi(\cdot|s)$ represents a valid probability distribution over actions. This policy family is required to satisfy a global expected entropy constraint:*

$$\mathbb{E}_{s\sim p(s)}[H(\pi^*(\cdot|s))] = \mathcal{H}_0^{global}, \tag{16}$$

*where $\mathcal{H}_0^{global}$ is a given constant.*

*Within the policy space $\Pi_{\mathcal{H}_0^{global}}$, the family of policies $\{\pi^*(\cdot|s)\}_{s\in\mathcal{S}}$ that maximizes the global expected action value $\mathbb{E}_{s\sim p(s)}[\mathbb{E}_{a\sim\pi(a|s)}[Q(s,a)]]$ has the property that, for each state $s$, the optimal policy $\pi^*(a|s)$ takes the form of a soft policy:*

$$\pi^*(a|s) = \frac{\exp(Q(s,a)/\alpha)}{\int_{a'\in\mathcal{A}} \exp(Q(s,a')/\alpha)da'}, \tag{17}$$

*where $\alpha > 0$ is a global temperature parameter, whose value is implicitly determined by a global expected entropy constraint: $\mathbb{E}_{s\sim p(s)}[H(\pi^*(\cdot|s))] = \mathcal{H}_0^{global}$.*

***Proof.*** We seek a family of policies $\{\pi(\cdot \mid s)\}_{s\in\mathcal{S}}$ belonging to the constrained space:

$$\Pi_{\mathcal{H}_0^{global}} = \left\{ \{\pi(\cdot \mid s)\}_{s\in\mathcal{S}} \,\middle|\, \mathbb{E}_{s\sim p(s)}\big[H(\pi(\cdot \mid s))\big] = \mathcal{H}_0^{global}, \int_{\mathcal{A}} \pi(a \mid s)\, da = 1, \, \forall s \right\}, \tag{18}$$

which maximises the expected action-value

$$J\big(\{\pi(\cdot \mid s)\}\big) = \mathbb{E}_{s\sim p(s)}\Big[\mathbb{E}_{a\sim\pi(\cdot|s)}[Q(s,a)]\Big] = \int_{\mathcal{S}} p(s)\int_{\mathcal{A}} \pi(a \mid s)\, Q(s,a)\, da\, ds. \tag{19}$$

Then, we introduce a scalar multiplier $\alpha$ for the global expected-entropy constraint and a state-dependent multiplier $\eta(s)$ for the normalisation constraint at each $s$. The Lagrangian reads

$$\mathcal{L}\big(\{\pi(\cdot \mid s)\}, \alpha, \{\eta(s)\}\big) = \int_{\mathcal{S}}\int_{\mathcal{A}} \Big[p(s)\pi(a \mid s)Q(s,a) - \alpha\, p(s)\pi(a \mid s)\log\pi(a \mid s) + \eta(s)\pi(a \mid s)\Big]\, da\, ds$$
$$- \alpha\, \mathcal{H}_0^{global} - \int_{\mathcal{S}} \eta(s)\, ds. \tag{20}$$

Because the decision variables for distinct states couple only through $\alpha$, we can minimise the integrand for each fixed $s$ independently:

$$\mathcal{L}_s\big(\pi(\cdot \mid s)\big) = \int_{\mathcal{A}} \Big[p(s)\pi(a \mid s)Q(s,a) - \alpha\, p(s)\pi(a \mid s)\log\pi(a \mid s) + \eta(s)\pi(a \mid s)\Big]\, da. \tag{21}$$

Taking the functional derivative and setting it to zero yields, for almost every $a \in \mathcal{A}$, we can obtain

$$p(s)Q(s,a) - \alpha\, p(s)\log\pi(a \mid s) - \alpha\, p(s) + \eta(s) = 0. \tag{22}$$

Assuming $p(s) > 0$, we divide both sides by $p(s)$ and rearrange:

$$\log\pi(a \mid s) = \frac{Q(s,a)}{\alpha} - 1 + \frac{\eta(s)}{\alpha p(s)}. \tag{23}$$

Let $\tilde{\eta}(s) = \eta(s)/p(s)$. Exponentiating gives the unnormalised form

$$\pi(a \mid s) = \exp\left(\frac{\tilde{\eta}(s) - \alpha}{\alpha}\right)\exp\left(\frac{Q(s,a)}{\alpha}\right) = C(s)\exp\left(\frac{Q(s,a)}{\alpha}\right), \tag{24}$$

where $C(s)$ is a state-wise normalising constant.

Imposing $\int_{\mathcal{A}} \pi(a \mid s) \, da = 1$, we can determine

$$C(s) = \Big[ \int_{\mathcal{A}} \exp\big(Q(s, a')/\alpha\big) \, da' \Big]^{-1}. \tag{25}$$

Therefore, the optimal policy family is the Boltzmann distribution

$$\pi^*(a \mid s) = \frac{\exp\big(Q(s, a)/\alpha\big)}{\displaystyle\int_{\mathcal{A}} \exp\big(Q(s, a')/\alpha\big) \, da'} \qquad \forall s \in \mathcal{S}, \ a \in \mathcal{A}. \tag{26}$$

The scalar $\alpha > 0$ is the Lagrange multiplier associated with the global entropy constraint and serves as a common temperature across all states. Its value is obtained implicitly by substituting $\pi^*$ back into

$$\mathbb{E}_{s \sim p(s)}\big[ H(\pi^*(\cdot \mid s)) \big] = \mathcal{H}_0^{\text{global}}. \tag{27}$$

Consequently, although the entropy constraint is imposed only on the state-averaged entropy, each per-state optimal policy still follows a Boltzmann form with the same temperature parameter $\alpha$.

## B  RELATED WORK

We review existing works on using the diffusion model as a policy function in combination with RL.

**Online RL with Diffusion Policy.**  Online RL enables agents to refine their policies through real-time interaction. Yang *et al.* introduced DIPO (Yang et al., 2023a), which maintains a dedicated diffusion buffer to store actions and model them using diffusion techniques. Psenka *et al.* proposed QSM (Psenka et al., 2023), which aligns policies with $\nabla_a Q$ via score matching, but is sensitive to value gradient inaccuracies across the action space. Recently, Ding *et al.* (Ding et al., 2024) proposed QVPO, which weights diffusion-sampled actions by Q-values without computing gradients. However, it uses a fixed ratio of uniform samples to boost the entropy, lacking adaptive control and later degrading performance. Ma *et al.* (Ma et al., 2025) proposed SDAC, which uses score matching over noisy energy-based diffusion. It avoids requiring optimal actions but suffers from high gradient variance due to poor sampling in high-$Q$ regions. Celik *et al.* proposed DIME (Celik et al., 2025), which derives a lower bound on the diffusion policy entropy and integrates it into the maximum-entropy RL framework. However, directly incorporating an inaccurate entropy estimate into the policy objective can degrade performance.

Complementary to these methods that train diffusion policies from scratch, a parallel line of work focuses on the online refinement of pre-trained diffusion policies. DPPO (Ren et al., 2024) formulates the reverse diffusion process as a secondary MDP and applies on-policy PPO-style optimization, achieving strong performance. Yuan *et al.* proposed Policy Decorator (Yuan et al., 2024), which treats a large base diffusion policy as a black box and learns a bounded residual policy with PPO to improve performance in a model-agnostic and stable manner. Ankile *et al.* introduced ResiP (Ankile et al., 2025), which regards a chunked imitation policy as a high-level planner and trains a closed-loop residual controller to provide fine-grained corrections for precise assembly. Wagenmaker *et al.* proposed DSRL (Wagenmaker et al., 2025), which steers a frozen diffusion policy by running RL in its latent noise space with a dual-$Q$ architecture, achieving sample-efficient online adaptation without finetuning the diffusion network weights.

**Offline RL with Diffusion Policy.**  Offline RL focuses on learning optimal policies from suboptimal datasets, with the core challenge being the out-of-distribution (OOD) problem (Kumar et al., 2020; Fujimoto et al., 2019). Diffusion models are naturally suited for offline RL due to their ability to model complex data distributions. Wang *et al.* proposed Diffusion-QL (Wang et al., 2023), which combines behavior cloning through a diffusion loss with Q-learning to improve policy learning. However, Diffusion-QL suffers from slow training and instability in OOD regions. To address the former, Kang *et al.* proposed Efficient Diffusion Policy (EDP) (Kang et al., 2023), which speeds up training by initializing from dataset actions and adopting a one-step sampling strategy. To mitigate OOD issues, Ada *et al.* introduced SRDP (Ada et al., 2024), which enhances generalization by integrating state reconstruction into the diffusion policy. Furthermore, Chen *et al.* proposed CPQL (Chen et al., 2023),

a consistency-based method that improves efficiency via one-step noise-to-action generation during both training and inference, albeit with some performance trade-offs. In parallel, Hansen-Estruch *et al.* proposed IDQL (Hansen-Estruch et al., 2023), which reinterprets IQL as a behaviour-regularised actor-critic method and uses a diffusion-model among the behaviour cloning policy to extract the implicit actor. Recently, Park *et al.* proposed Flow Q-Learning (FQL) (Park et al., 2025), which leverages an expressive flow-matching policy together with a separately RL-trained one-step actor to model complex action distributions without backpropagating through iterative generation, achieving competitive results across large-scale offline and offline-to-online benchmarks.

**Diffusion Acceleration.** The pursuit of efficient diffusion sampling has yielded several key advancements. Denoising Diffusion Implicit Models (DDIM) (Song et al., 2020a) first re-envisioned the reverse process as a deterministic ODE, permitting significant sampling speed-ups. DPM-Solver (Lu et al., 2022) introduced high-order exponential integrators, achieving high-fidelity generation without retraining. DPM-Solver++ (Lu et al., 2025a) further adapted this high-order approach for the widely-used classifier-free guidance regime, stabilizing sampling at large guidance scales. Concurrently, Consistency Models (Song et al., 2023) explored a distillation-based approach, compressing the multi-step ODE trajectory into a single "consistency function" that maps noise to data in one or few steps.

**Comparison with DACER.** Wang *et al.* proposed DACER (Wang et al., 2024), which leverages the reverse diffusion process as a policy approximator and employs a Gaussian Mixture Model (GMM) to estimate entropy for balancing exploration and exploitation. However, this approach lacks a theoretical justification for how maximizing the expected Q-value under entropy regularization inherently fosters multimodal policies when using diffusion models as policy functions. Furthermore, DACER remains constrained by a critical trade-off: while long diffusion processes ensure high performance, they severely hinder training efficiency; conversely, reducing steps leads to performance degradation. In contrast, our method, DACERv2, resolves this bottleneck by introducing a Q-gradient field objective, incorporated with a time-weighted mechanism and Q-gradient normalization. These innovations enable valid policy approximation with significantly fewer diffusion steps, thereby improving efficiency while maintaining or even improving both performance and policy multimodality.

**Comparison with QSM.** Psenka *et al.* proposed QSM (Psenka et al., 2023), an algorithm that aligns diffusion model policies with $\nabla_a Q(s, a)$ by leveraging their score-based structure. Both methods leverage Q-gradients for diffusion policy optimization. QSM employs score matching, whereas DACERv2 performs end-to-end Q-value maximization augmented with a time-weighted score-matching loss and entropy regularization, resulting in a multi-task objective. DACERv2 additionally stabilizes Q-gradients through normalization and improves efficiency, converging in just 5 diffusion steps compared to QSM's approximately 20.

## C ENVIRONMENTAL DETAILS

**MuJoCo** (Brockman et al., 2016): This is a high-performance physics simulation platform widely adopted for robotic reinforcement learning research. The environment features efficient physics computation, accurate dynamic system modeling, and comprehensive support for articulated robots, making it an ideal benchmark for RL algorithm development. In this research, we concentrate on eight tasks: Humanoid, Ant, HalfCheetah, Walker2d, InvertedDoublePendulum (IDP), Hopper, HumanoidStandup, and Swimmer. The IDP task entails maintaining the balance of a double pendulum in an inverted state. In contrast, the objective of the other tasks is to maximize the forward velocity while avoiding falling. All these tasks are realized through the OpenAI Gym interface (Brockman, 2016).

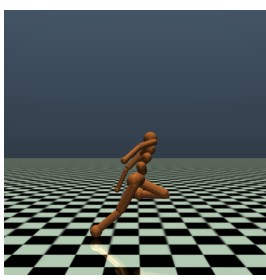

*State-action space*: $\mathcal{S} \in \mathbb{R}^{17}, \mathcal{A} \in \mathbb{R}^6$.

***Objective.*** Maintain forward velocity as fast as possible while avoiding falling over.

***Initialization.*** The walker is initialized in a standing position with slight random noise added to joint positions and velocities.

***Termination.*** The episode ends when the agent falls, the head touches the ground, or after 1000 steps.

Figure 5: Walker2d-v3

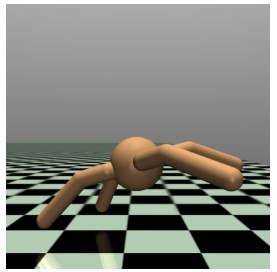

*State-action space*: $\mathcal{S} \in \mathbb{R}^{376}, \mathcal{A} \in \mathbb{R}^{17}$.

***Objective.*** Maintain balance and walk or run forward at a high velocity while avoiding falls.

***Initialization.*** The humanoid starts in an upright position with slight random perturbations to joint angles and velocities.

***Termination.*** The episode ends when the head height is less than 1.0 meter, the torso tilts excessively, or after 1000 steps.

Figure 6: Humanoid-v3

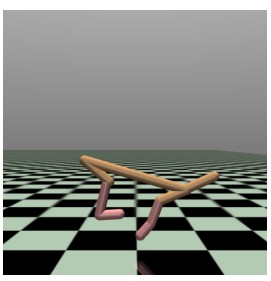

*State-action space*: $\mathcal{S} \in \mathbb{R}^{111}, \mathcal{A} \in \mathbb{R}^8$.

***Objective.*** Navigate forward as quickly as possible using four legs while maintaining stability.

***Initialization.*** The ant is initialized in a stable, upright position with random noise applied to its joints.

***Termination.*** The episode ends if the ant falls, flips over, or reaches the maximum step count of 1000.

Figure 7: Ant-v3

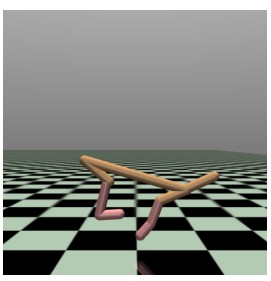

*State-action space*: $\mathcal{S} \in \mathbb{R}^{17}, \mathcal{A} \in \mathbb{R}^6$.

***Objective.*** Achieve maximum forward velocity with smooth, coordinated movements.

***Initialization.*** The agent starts with a slight forward tilt and randomized joint noise.

***Termination.*** The episode ends after 1000 steps or if the agent's head touches the ground.

Figure 8: Halfcheetah-v3

## D  EXPERIMENTAL HYPERPARAMETERS

The hyperparameters of all baseline algorithms except the diffusion-based algorithm are shown in Table 4. Additionally, the parameters for all diffusion-based algorithms, including DACERv2, are presented in Table 5 and Table 6.

The hyperparameter $c, d$ for time-weighted mechanism is determined by the diffusion step size, inspired by the variance-preserving beta schedule used in DDPM (Ho et al., 2020). The code of implementation is as follows:

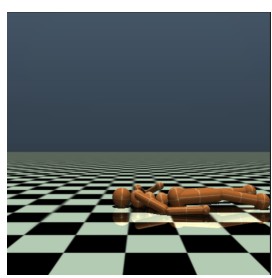

Figure 9: Swimmer-v3

*State-action space*: $\mathcal{S} \in \mathbb{R}^8, \mathcal{A} \in \mathbb{R}^2$.

*Objective.* Propel forward through water-like dynamics using sinusoidal wave patterns.

*Initialization.* The swimmer starts in a straight posture with minor random perturbations.

*Termination.* The episode ends after 1000 steps, with no explicit termination for falling.

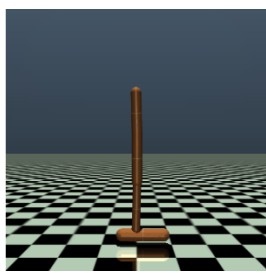

Figure 10: Humanoid-Standup

*State-action space*: $\mathcal{S} \in \mathbb{R}^{348}, \mathcal{A} \in \mathbb{R}^{17}$.

*Objective.* Stand up from lying on the ground by applying torques to the joints, with rewards for upward movement and penalties for large actions or strong impacts.

*Initialization.* The humanoid starts lying down, with small random noise added to joint positions and velocities.

*Termination.* The episode does not terminate early; it ends after 1000 steps.

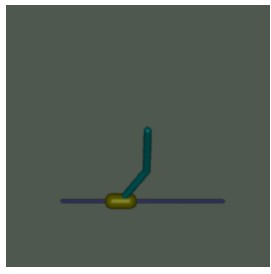

Figure 11: Hopper-v3

*State-action space*: $\mathcal{S} \in \mathbb{R}^{11}, \mathcal{A} \in \mathbb{R}^3$.

*Objective.* Hop forward as fast as possible by applying torques to the thigh, leg, and foot joints, while staying upright.

*Initialization.* The hopper starts standing upright with small random perturbations in position and velocity.

*Termination.* The episode ends if the hopper falls (body hits the ground) or after 1000 steps.

Figure 12: IDP-v3

*State-action space*: $\mathcal{S} \in \mathbb{R}^9, \mathcal{A} \in \mathbb{R}^1$.

*Objective.* Balance the second pole upright by applying horizontal forces to the cart, while maximizing time alive and minimizing tip distance and joint velocities.

*Initialization.* The cart and poles start near the upright position with small random noise in position and velocity.

*Termination.* The episode ends if the tip of the second pole falls below height 1. Otherwise, it is truncated after 1000 steps.

```python
def vp_alpha_schedule(timesteps: int, b_min=0.1, b_max=10.):
    T = timesteps
    t = np.arange(1, T + 1)
    return np.exp(-b_min / T - 0.5 * (b_max - b_min) * (2 * t - 1) / T ** 2)

# Set parameters
timesteps = 5
alphas = vp_alpha_schedule(timesteps)
```

```
# Reverse the alpha array as in B.alphas[self.agent.num_timesteps - 1 - t]
reversed_alphas = alphas[::-1]
t_vals = np.arange(timesteps)

# Fit the exponential form exp(ct + d)
params, _ = curve_fit(exp_fit, t_vals, reversed_alphas)
c, d = params
```

TABLE 4
BASELINE HYPERPARAMETERS.

| Hyperparameters | Value |
|---|---|
| *Shared* | |
| Replay buffer capacity | 1,000,000 |
| Buffer warm-up size | 30,000 |
| Batch size | 256 |
| Action bound | $[-1, 1]$ |
| Hidden layers in critic network | [256, 256, 256] |
| Hidden layers in actor network | [256, 256, 256] |
| Activation in critic network | GeLU |
| Activation in actor network | GeLU |
| Optimizer | Adam ($\beta_1 = 0.9, \beta_2 = 0.999$) |
| Actor learning rate | 1e−4 |
| Critic learning rate | 1e−4 |
| Discount factor ($\gamma$) | 0.99 |
| Policy update interval | 2 |
| Target smoothing coefficient ($\rho$) | 0.005 |
| Reward scale | 0.2 |
| *Maximum-entropy framework* | |
| Learning rate of $\alpha$ | 3e−4 |
| Expected entropy ($\overline{\mathcal{H}}$) | $\overline{\mathcal{H}} = -\dim(\mathcal{A})$ |
| *Deterministic policy* | |
| Exploration noise | $\epsilon \sim \mathcal{N}(0, 0.1^2)$ |
| *Off-policy* | |
| Replay buffer size | $1 \times 10^6$ |
| Sample batch size | 20 |
| *On-policy* | |
| Sample batch size | 2,000 |
| Replay batch size | 2,000 |

TABLE 5
HYPERPARAMETER $\eta$ USED IN DACERv2.

| **Task** | Hopper | Ant | HalfCheetah | Walker2d | MultiGoal | Hum. S. | Humanoid | Swimmer | IDP |
|---|---|---|---|---|---|---|---|---|---|
| $\eta$ | 1.0 | 1.0 | 1.0 | 1.0 | 1.0 | 0.01 | 0.01 | 0.01 | 0.01 |

# E  LIMITATION AND FUTURE WORK

In this study, we propose the Q-gradient field objective as an auxiliary training loss to provide more informative gradient signals for guiding the diffusion policy. However, algorithms such as PPO (Schulman et al., 2017) and GRPO (Shao et al., 2024) do not explicitly learn a Q-function, making it challenging to directly integrate the diffusion policy of DACERv2 and its associated loss function with these methods. This indicates that the generality of our method is currently affected by the presence of value functions. Future work could explore reformulating the auxiliary objective into a purely trajectory-based form, thereby enabling integration with methods that rely solely on policy gradients.

TABLE 6
DIFFUSION-BASED ALGORITHMS' HYPERPARAMETERS

| Parameter | DACERv2 | DACER | QVPO | QSM | DIME | DIPO |
|---|---|---|---|---|---|---|
| Replay buffer capacity | 1e6 | 1e6 | 1e6 | 1e6 | 1e6 | 1e6 |
| Buffer warm-up size | 3e4 | 3e4 | 3e4 | 3e4 | 3e4 | 3e4 |
| Batch size | 256 | 256 | 256 | 256 | 256 | 256 |
| Discount $\gamma$ | 0.99 | 0.99 | 0.99 | 0.99 | 0.99 | 0.99 |
| Target network soft-update rate $\rho$ | 0.005 | 0.005 | 0.005 | 0.005 | N/A | 0.005 |
| Network update times per iteration | 1 | 1 | 1 | 1 | 1 | 1 |
| Action bound | $[-1, 1]$ | $[-1, 1]$ | $[-1, 1]$ | $[-1, 1]$ | $[-1, 1]$ | $[-1, 1]$ |
| Reward scale | 0.2 | 0.2 | 0.2 | 0.2 | 0.2 | 0.2 |
| No. of Actor layers | 2 | 2 | 2 | 2 | 2 | 2 |
| No. of Actor hidden dims | 256 | 256 | 256 | 256 | 256 | 256 |
| No. of Critic layers | 2 | 2 | 2 | 2 | 2 | 2 |
| No. of Critic hidden dims | 256 | 256 | 256 | 256 | 2048 | 256 |
| Activations in critic network | GeLU | GeLU | Mish | ReLU | ReLU | Mish |
| Activations in actor network | Mish | Mish | Mish | ReLU | ReLU | Mish |
| **Diffusion steps** | **5** | 20 | 20 | 20 | 16 | 20 |
| Policy delay update | 2 | 2 | 2 | 2 | 2 | 2 |
| Action gradient steps | N/A | N/A | N/A | N/A | N/A | 20 |
| No. of Gaussian distributions | 3 | 3 | N/A | N/A | N/A | N/A |
| No. of action samples | 200 | 200 | N/A | N/A | N/A | N/A |
| Time-weighted hyperparameter $c$ | 0.4 | N/A | N/A | N/A | N/A | N/A |
| Time-weighted hyperparameter $d$ | -1.8 | N/A | N/A | N/A | N/A | N/A |
| Alpha delay update | 10,000 | 10,000 | N/A | N/A | N/A | N/A |
| Noise scale $\lambda$ | 0.1 | 0.1 | N/A | N/A | N/A | N/A |
| Optimizer | Adam | Adam | Adam | Adam | Adam | Adam |
| Actor learning rate | $1 \cdot 10^{-4}$ | $1 \cdot 10^{-4}$ | $1 \cdot 10^{-4}$ | $1 \cdot 10^{-4}$ | $3 \cdot 10^{-4}$ | $1 \cdot 10^{-4}$ |
| Critic learning rate | $1 \cdot 10^{-4}$ | $1 \cdot 10^{-4}$ | $1 \cdot 10^{-4}$ | $1 \cdot 10^{-4}$ | $3 \cdot 10^{-4}$ | $1 \cdot 10^{-4}$ |
| Alpha learning rate | $3 \cdot 10^{-2}$ | $3 \cdot 10^{-2}$ | N/A | N/A | $1 \cdot 10^{-3}$ | N/A |
| Target entropy | $-\dim(\mathcal{A})$ | $-\dim(\mathcal{A})$ | N/A | N/A | $-4\dim(\mathcal{A})$ | N/A |

# F EXTRA ABLATION STUDY

We conducted an ablation study on the Humanoid-v3 task to examine the effect of normalizing the Q-gradient. The results presented in Fig. 13 demonstrate that normalization method consistently enhance performance returns.

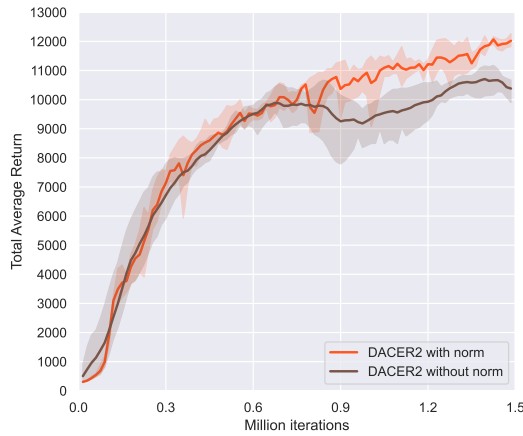

Figure 13: Ablation on the normalization of Q-function.

## G  LLM STATEMENT

Large Language Models (LLMs) were employed solely for language refinement in this paper. Specifically, we used them to polish grammar, improve clarity, and enhance the academic style of our writing. The role of LLMs was limited to editing and improving the presentation of the text, without contributing to the technical content.

