# OpenReview forum: "Enhanced DACER Algorithm with High Diffusion Efficiency"
_ICLR.cc/2026/Conference — Submitted to ICLR 2026_

### Official Review · Reviewer_dk8y · 2025-10-29

**Soundness:** 3
**Presentation:** 3
**Contribution:** 2
**Rating:** 6
**Confidence:** 4

**Summary:**

This paper proposes DACERv2, an enhanced version of the diffusion-based online reinforcement learning algorithm, DACER. The authors identify a key limitation in DACER: a trade-off between performance and efficiency, where a large number of diffusion steps is required for high performance, leading to high computational cost.
To address this, DACERv2 introduces two main contributions:
1. **A Q-Gradient Field Objective ($\mathcal{L}_g$)**: This is an auxiliary loss term that provides intermediate supervision at each step $t$ of the denoising process. It is motivated by the connection between the optimal soft-Q policy (a Boltzmann distribution) and the score function via Langevin dynamics, aiming to align the policy's score function $S_{\theta}(s, a_t, t)$ with the normalized Q-gradient $\nabla_{a_t} Q(s, a_t)$.
2. **A Temporal Weighting Mechanism ($w(t)$)**: This mechanism modulates the strength of the Q-gradient objective based on the diffusion timestep $t$. This is designed to resolve the inconsistency between the time-independent Q-gradient field and the time-dependent nature of the diffusion denoising process.
The authors claim that this new combined objective ($\mathcal{L}_\pi = \mathcal{L}_q + \eta \mathcal{L}_g$) allows DACERv2 to achieve state-of-the-art (SOTA) performance on complex MuJoCo benchmarks using only 5 diffusion steps. This results in significant improvements in both training and inference efficiency.

**Strengths:**

1. **Clear and Significant Problem**: The paper addresses a critical, practical limitation of diffusion policies—their poor computational efficiency due to the high number of sampling steps.
2. **Strong Empirical Results**: The primary claim of achieving SOTA performance with only $T=5$ steps is backed by comprehensive experiments. The efficiency gains shown in Table 1 are dramatic and highly compelling (e.g., >3.5x faster inference than DACER). The ablations in Figure 4 clearly isolate the impact of the two key contributions ($\mathcal{L}_g$ and $w(t)$), empirically validating their necessity for the observed performance.

**Weaknesses:**

1. **Lack of Theoretical Novelty and Justification**: The paper's theoretical support is weak on two fronts.
    - **Heuristic Contribution**: The paper's primary novel contribution, the $\mathcal{L}_g$ auxiliary loss, is a pure heuristic. It is motivated by analogy but lacks any formal proof or analysis showing that the combined objective ($\mathcal{L}_\pi = \mathcal{L}_q + \eta \mathcal{L}_g$) leads to a better, faster, or more stable convergence to the optimal policy.
    - **Non-Novel Theorem**: The main theoretical result presented (Theorem 1 in Appendix A) appears to be a restatement of a standard, known result (i.e., that maximizing value under a global entropy constraint yields a Boltzmann policy with a global temperature). This theorem only justifies the baseline DACER objective and does not represent a novel contribution of this work.
2. **Potentially High Hyperparameter Sensitivity ($\eta$)**: The hyperparameter $\eta$ (the auxiliary loss weight) is clustered into two groups (Table 5), but the values are $1.0$ and $0.01$—a 100-fold difference. This implies that the algorithm's performance is highly sensitive to this choice. The paper offers no insight into what task properties (e.g., dimensionality) necessitate such a drastic change, which is crucial for applying this method to new environments.

**Questions:**

1. Regarding the hyperparameter $\eta$ in Table 5: The optimal value differs by 100x ($1.0$ vs. $0.01$) across tasks. What properties of the environment (e.g., dimensionality, task complexity) dictate this choice? How sensitive is the algorithm to this parameter?
2. The paper's main theoretical support, Theorem 1, appears to be a standard result for justifying the soft-Q objective. Given this, can the authors provide any novel theoretical analysis for their actual contribution, $\mathcal{L}_g$? For instance, can it be shown that $\mathcal{L}_g$ acts as a variance reduction term, or that the combined objective $\mathcal{L}_\pi$ has superior convergence properties compared to optimizing $\mathcal{L}_q$ alone?

---

> ### Author Response · Authors · 2025-11-21
> **Re: the Reviewer dk8y**
>
> We thank you for the careful reading of our paper and constructive comments in detail. Below are our responses to your concerns.
>
> # > Weakness1
> We clarify that Theorem 1 (Appendix A) is a standard result, **not** a novel contribution of this work. We included it (as referenced in the preliminary section) solely to formally establish the target Boltzmann distribution as the optimal policy—a conclusion not explicitly drawn in the original DACER. This addition provides the foundation for DACERv2.
>
> Our primary contribution is a new training objective that addresses a fundamental inference inefficiency in the original DACER. The core issue is **sparse supervision**: the original DACER receives a Q-function learning signal only at the final denoising step ($t$=0 of 20). Consequently, gradients back-propagated to earlier diffusion steps become **increasingly indirect and attenuated**, analogous to the vanishing gradients in RNNs [1]. This makes it difficult for the diffusion policy to learn to approximate the Boltzmann distribution of the Q-function. To resolve this, we introduce **intermediate supervision** at every diffusion step. Our approach leverages the insight that the optimal policy in the original DACER (derived from maximizing the Q-value under an entropy constraint) and the policy induced by Langevin dynamics share the same Boltzmann form. By implementing this intermediate supervision step-wise, the policy receives a significantly denser and more direct learning signal.
>
> To ensure stable training of this joint objective (terminal plus intermediate supervision), **we developed a time-weighting mechanism and a Q-normalization method.** As a result, our method achieves SOTA performance—demonstrating enhanced multi-modality and sample efficiency—along with significant acceleration. Compared to the original DACER, our method reduces training and inference times by **41.7%** and **60.6%**, respectively.
>
> In summary, the motivation behind our method extends beyond a simple combination of loss functions; rather, it aims to replace the purely distal supervision used in prior algorithms with local, step-wise guidance constructed through auxiliary objectives. This introduces a proximal supervision signal at each denoising step, intended to foster a monotonic and efficient refinement process. Consequently, intermediate actions tend to align progressively with the target Boltzmann distribution, facilitating faster convergence
>
>
> # > Weakness2 & Question1
> $\eta$ is a hyperparameter that requires tuning. We recommend a logarithmic-scale search, starting from 1.0 and decaying by factors of 10. As you noted, understanding the sensitivity of $\eta$ is important. In our ablation study (Section 4.4) on the Humanoid-v3 task, we found that performance is sensitive **to order-of-magnitude changes** but remains robust to smaller variations **within the same order of magnitude**.
>
> # > Question2
> Thank you for your suggestion, we demonstrate that under specific assumptions, the Signal-to-Noise Ratio (SNR) of the gradients is higher when incorporating $\mathcal{L}_g$ into the objective function, compared to using $\mathcal{L}_q$ alone. A higher SNR implies more stable policy updates. The detailed proof is provided in the [link](https://ibb.co/W4Z04VKc).
>
> # > Reference
>
> [1] Pascanu, Razvan, Tomas Mikolov, and Yoshua Bengio. "On the difficulty of training recurrent neural networks." International conference on machine learning. Pmlr, 2013.

---

> ### Author Response · Authors · 2025-11-26
>
> Thank you again for the great efforts and valuable comments. We have carefully addressed the main concerns in detail. We hope you might find the response satisfactory. As the discussion phase is about to close, we are very much looking forward to hearing from you about any further feedback. We will be very happy to clarify any further concerns (if any).

---

### Official Review · Reviewer_VwAt · 2025-10-30

**Soundness:** 1
**Presentation:** 3
**Contribution:** 1
**Rating:** 2
**Confidence:** 5

**Summary:**

This paper introduces DACERv2, an improved version of DACER that learns a Q-gradient field for fast action denoising. Observing that the denoising process in diffusion relies on a time-dependent score function, DACER-v2 also introduces such dependency by scaling the Q-gradient with a crafted exponential decay function. Finally, DACER-v2 is evaluated on tasks from Gym-MuJoCo and demonstrates improved performance as compared to several diffusion policy baselines.

**Strengths:**

The overall idea is clearly presented and straightforward to implement. The proposed method is efficient both in terms of training and inference, which makes it preferable for deployment in embodied scenarios.

**Weaknesses:**

The idea of aligning the score networks with the gradient of Q-value functions has been extensively investigated in QSM [1], DAC [2], QGPO [3], iDEM [4], and [5]. One contribution of DACER-v2 seems to be the time-based weighting. However, this is purely heuristic and theoretically unjustified. On the other hand, QGPO, iDEM, and [5] also estimate the time-dependent score, and their estimations are exact in theory. Therefore, the novelty and insight of this paper are limited.

Besides, this paper lacks a related work section to familiarize the readers with the frontier literature. For example, given that the proposed algorithm is termed DACER-v2, it is necessary to include detailed introductions about DACER and demonstrate how v2 improves the v1 algorithm.

[1] Learning a Diffusion Model Policy from Rewards via Q-Score Matching.

[2] Diffusion Actor-Critic: Formulating Constrained Policy Iteration as Diffusion Noise Regression for Offline Reinforcement Learning.

[3]: Contrastive Energy Prediction for Exact Energy-Guided Diffusion Sampling in Offline Reinforcement Learning.

[4]: Iterated Denoising Energy Matching for Sampling from Boltzmann Densities.

[5]: Sampling from Energy-based Policies using Diffusion.

**Questions:**

How many environment frames/steps does one iteration correspond to?

The authors mentioned that the Q-gradient prediction is an auxiliary objective (line 208). However, I don’t see any further introduction about the actual objective of the diffusion policy in the paper. Could the authors make this clear?

---

> ### Author Response · Authors · 2025-11-21
> **(1/2) Re: the Reviewer VwAt**
>
> We appreciate your detailed feedback. Below are our responses to your concerns.
>
> # > Weakness1
> We apologize for any ambiguity in our initial presentation. We wish to clarify that our motivational starting point fundamentally differs from that of the cited works. The distinctions between DACERv2 and these methods are detailed below:
>
> ## 1. Comparison with Diffusion-based Online RL
> *   **QSM [1]:** Although QSM also aligns diffusion scores with Q-gradients, it relies on **pure Q-score matching within an online RL setting, requiring around a 20-step sampler.** This approach is known to suffer from high variance and unstable optimization. In contrast, DACERv2 does not employ the Q-gradient as the primary objective. Instead, we retain the robust max-Q objective from the original DACER and utilize the Q-gradient as an auxiliary signal, carefully adapted via temporal weighting and normalization. This design effectively remedies the inefficiencies of raw Q-gradient application, enabling DACERv2 to achieve high performance with only 5 diffusion steps—an efficiency level QSM cannot match. Thus, despite the shared use of Q-gradient guidance, **the objectives, training dynamics, and practical outcomes of the two methods are fundamentally different.**
>
> *   **DQS [5]:** DQS formulates the policy explicitly as a energy-based policy $\pi(a|s) \propto \exp(Q(s,a))$ and proposes a diffusion-based sampler, together with an actor–critic training procedure, to sample from and learn this energy-based policy in continuous action spaces. While this improves the faithfulness of energy-based sampling, it still relies on multi-step denoising typical of diffusion samplers, which can be computationally heavy for real-time online interaction. Conversely, DACERv2 is designed not to reconstruct the Q-distribution perfectly, but to optimize the trade-off between computational efficiency and effective policy improvement. We demonstrate that our specific weighting mechanism extracts sufficient Q-gradient signals to guide the policy with as few as 5 steps—a practical breakthrough in efficiency that DQS does not primarily address.
>
> ## 2. Comparison with Diffusion-based Offline RL
> *   **DAC [2]:** Designed for offline RL (e.g., D4RL), DAC employs a diffusion model within a KL-constrained policy iteration framework to **adhere closely to a fixed behavior policy.** A soft Q-guidance term is used solely to prevent out-of-distribution (OOD) actions. In contrast, DACERv2 targets online RL, where no fixed behavior policy exists. Here, the diffusion model serves as the primary control policy rather than a regularizer. The Q-gradient field (augmented by temporal weighting and normalization) functions purely as an auxiliary denoising signal to enhance the efficiency and stability of the reverse process under a minimal step count.
>
> *   **QGPO [3]:** While QGPO is a significant step in combining diffusion models with Q-functions, it differs conceptually. QGPO, derived from the CEP framework, **is an offline method assuming a fixed dataset and a stable energy function**, aiming to learn an exact time-dependent energy-guidance field. DACERv2, however, is an online algorithm where both policy and Q-function are continuously updated. QGPO's theoretical assumptions cannot be directly applied to our setting, which must contend with distribution shifts and a constantly evolving Q-function.
>
> ## 3. Comparison with Diffusion-based Samplers
> *   **iDEM [4]:** A fundamental difference exists in problem formulation. iDEM is designed as a sampler for a **fixed, known density (static energy function $E(x)$)**, assuming $E(x)$ is the immutable "ground truth." It minimizes Fisher Divergence to match this distribution exactly. DACERv2 operates in a fully Online RL setting where the Q-function is a moving target—a function approximator that is continuously updating and initially exhibits high variance and bias. Forcing the policy to perfectly match the gradient of a non-converged, noisy Q-function (as iDEM's theory would suggest) would lead to overfitting and instability. While iDEM solves sampling from a static oracle, our method addresses policy optimization under a dynamic, noisy objective.

---

> ### Author Response · Authors · 2025-11-21
> **(2/2) Re: the Reviewer VwAt**
>
> ## Clarification on Novelty and Insights
> The reviewer questions the novelty of our work, characterizing our time-based weighting as "heuristic" compared to the "theoretically exact" estimations in QGPO or iDEM. We respectfully argue that this distinction embodies the core insight of our paper, tailored specifically for the unique challenges of Online RL. In offline or static sampling settings, striving for an exact match to a fixed function is theoretically sound. However, in Online RL, the Q-function is a Moving Target characterized by significant estimation errors during training. Applying an "exact theoretical estimation" to an inaccurate Q-function merely amplifies bias, leading to error propagation and training instability.
>
> Furthermore, DACERv2 is designed to optimize the trade-off between efficiency and performance. While the original DACER theoretically ensures convergence to the Boltzmann distribution, its reliance on max-Q optimization causes back-propagated gradients to **become increasingly indirect and attenuated over long chains, akin to the vanishing gradient problem in RNNs [6].** This attenuation results in suboptimal diffusion efficiency. By introducing Q-gradients as intermediate supervisory signals—complemented by our time-weighting and normalization mechanisms—we not only improve algorithmic stability but significantly enhance efficiency, achieving superior results with fewer denoising steps compared to the baseline.
>
> # > Weakness2
> In the introduction, we kindly note that we have already provided a categorization and summary of existing approaches that integrate diffusion models with online RL, and we sincerely appreciate the reviewer’s attention to this part. We also point out that **sufficient consideration of how to balance diffusion efficiency and performance is still lacking** in current approaches. Due to space limitations, a comprehensive related work section is provided in Appendix B of our manuscript. This section details relevant literature in both Online RL and Offline RL, with a particular focus on their integration with Diffusion Policies. We also specifically compared the differences between the original DACER and DACERv2.
>
> The final paragraph of the introduction section summarizes our core contributions. To enhance the original DACER's diffusion efficiency, DACERv2 innovates within the training process itself. It introduces an additional Q-gradient field objective on top of the original DACER objective function. To achieve higher algorithmic performance, we further designed a time-weighting mechanism and introduced a Q-normalization method to ensure stability. This combined approach enables DACERv2 to achieve superior performance and stronger multi-modal action distributions using only 5 diffusion steps.
>
> # > Question1
> For the sake of fairness in comparison, we maintained the same experimental setup and plotting standards as used in DSAC[7] and DACER. In DSAC and DACER, **each iteration represents one network update, and 20 samples are sampled per iteration**.
>
> # > Question2
> Reviewer may kindly refer to Method Section 3.3 (lines 261–269), where the objective of the diffusion policy is explicitly described. We sincerely encourage the reviewer to revisit this part for clarification, and we are very happy to address any further questions or concerns.
>
> # > Reference
> [1] Psenka, Michael, et al. "Learning a diffusion model policy from rewards via q-score matching." arXiv preprint arXiv:2312.11752 (2023).
>
> [2] Fang, Linjiajie, et al. "Diffusion actor-critic: Formulating constrained policy iteration as diffusion noise regression for offline reinforcement learning." arXiv preprint arXiv:2405.20555 (2024).
>
> [3] Lu, Cheng, et al. "Contrastive energy prediction for exact energy-guided diffusion sampling in offline reinforcement learning." International Conference on Machine Learning. PMLR, 2023.
>
> [4] Akhound-Sadegh, Tara, et al. "Iterated denoising energy matching for sampling from boltzmann densities." arXiv preprint arXiv:2402.06121 (2024).
>
> [5] Jain, Vineet, Tara Akhound-Sadegh, and Siamak Ravanbakhsh. "Sampling from energy-based policies using diffusion." arXiv preprint arXiv:2410.01312 (2024).
>
> [6] Pascanu, Razvan, Tomas Mikolov, and Yoshua Bengio. "On the difficulty of training recurrent neural networks." International conference on machine learning. Pmlr, 2013.
>
> [7] Duan, Jingliang, et al. "Distributional soft actor-critic with three refinements." IEEE Transactions on Pattern Analysis and Machine Intelligence (2025).

---

> > ### Comment · Reviewer_VwAt · 2025-11-23
> >
> > I appreciate the authors' effort in providing a detailed comparison between DACERv2 and the listed methods. While I recognize that finding an approach exactly identical to DACERv2 is unlikely, the core principle—namely, aligning the policy's output with the gradient of the Q-function—is a well-established and extensively investigated idea in both the offline and online reinforcement learning literatures. QSM (in online RL) and DAC (in offline RL) serve as prominent representatives of this line of research. Furthermore, subsequent works like DQS, QGPO, and iDEM have significantly advanced this concept by offering exact estimations of the annealed Q-value function, which is crucial for mitigating the slow-mixing issues associated with Langevin dynamics; while the time-dependent weighting in DACERv2 simply decays the gradient and remains theoretically unclear. From this perspective, the novelty of DACERv2 appears to be primarily limited to a combination of this pre-existing gradient-alignment technique with the original DACER framework.
> >
> > Given that the rebuttal resolved my rest questions, I will increase my evaluation to 4.

---

> > > ### Author Response · Authors · 2025-12-03
> > >
> > > We thank the reviewer for their in-depth knowledge of the Offline RL literature. However, we respectfully point out that conflating the objectives of Offline RL with the distinct challenges of Online RL leads to a misunderstanding of our core contribution.
> > >
> > > ## 1. Offline RL vs. Online RL
> > > We respectfully argue that conflating Offline RL objectives with the real-time constraints of Online RL obscures the core contribution of DACERv2. While Q-gradient alignment has been extensively explored in Offline RL, these methods operate in a setting where inference latency is not a limiting factor; multi-step iterative refinement is acceptable when no real-time interaction loop is involved.
> > >
> > > In Online RL, however, each policy evaluation is subject to strict latency constraints. Although QSM introduced Q-guided score correction to this setting, it fails to address the central challenge of maintaining policy quality under severely reduced diffusion steps. **It requires around 20 steps to achieve viable performance, yet still falls significantly short of the original DACER and DACERv2.** This leaves a significant performance-efficiency gap in the Online Diffusion RL paradigm.
> > >
> > > To the best of our knowledge, DACERv2 is the first method to explicitly address this challenge by leveraging the theoretical consistency between the Boltzmann form of the optimal policy and Langevin dynamics parameterized by value networks. **This formulation provides denser learning signals, enabling approximate step-wise gradient descent for the monotonic refinement of generated actions.** By achieving superior performance with as few as 5 reverse-diffusion steps, DACERv2 satisfies the real-time requirements of online interaction—a capability unmatched by either prior offline methods or QSM.
> > >
> > > ## 2. Slow-mixing issue & Time-weighted mechanism
> > > It is important to distinguish our framework from settings where slow mixing is a bottleneck. The slow-mixing issue is inherent to non-annealed Langevin dynamics requiring MCMC sampling to traverse multimodal energy landscapes. **Our method, however, operates as a finite-step reverse diffusion process defined by the diffusion SDE (eg. DDPM) formulation**. The Q-gradient serves only to guide the generation within this fixed trajectory rather than driving an MCMC chain to convergence. Thus, the slow-mixing issue is not applicable to our approach.
> > >
> > > Regarding the time-weighted mechanism, it is designed to dynamically adjust the Q-gradient's influence based on step $t$, balancing early-stage guidance with late-stage refinement. This modulation strategy draws theoretical inspiration from the Variance Preserving (VP) schedule in DDPMs, rendering the design logical and consistent with established diffusion theory. **Moreover, our ablation studies empirically confirm that this mechanism is critical for achieving optimal performance.**
> > >
> > > In summary, DACERv2 addresses the critical balance between performance and efficiency in diffusion-based online RL. Crucially, our contribution extends beyond the mere addition of Q-gradients to the original DACER framework.

---

### Official Review · Reviewer_n3U9 · 2025-11-01

**Soundness:** 2
**Presentation:** 3
**Contribution:** 2
**Rating:** 4
**Confidence:** 4

**Summary:**

This paper proposes DACERv2, an online RL algorithm built upon DACER and employs a new Q-gradient field matching objective in the policy learning loss. The proposed method enables action sampling with five diffusion steps and outperforms the included baselines on OpenAI Gym environments. The authors also introduce a temporal weighting function that adjusts gradient magnitude across diffusion timesteps.

**Strengths:**

1. The proposed algorithm achieves strong performance on state-based OpenAI Gym environments, with higher training and inference efficiency than most baselines.
2. The paper is well-written.

**Weaknesses:**

1. The score function in a standard diffusion SDE is the score function of the perturbed distribution $\int q_{t|0}(a_t|a_0) \frac{e^{\frac{1}{\alpha}Q(s, a_0)}}{Z(s)}da_0$ and is not in the form of Equation (9). Moreover, the non-annealed Langevin dynamics used in this paper may suffer from slow mixing, as shown in [1].
2. The method proposed in this paper is a straightforward combination of the QSM [2] policy training loss (with a newly introduced weighting function) and the DACER policy training loss, and the analysis is insufficient to explain why this combination boosts performance without gradient-conflict issues.
3. The argument in Lines 698-699 is not well supported. If the optimal action with the largest Q value is a single point, then maximizing the Q-value will result in a delta distribution, not a multimodal policy. The multimodal property is more likely due to the entropy regularization and the expressive capacity of diffusion models.

[1] Song Y, Ermon S. Generative modeling by estimating gradients of the data distribution[J]. Advances in neural information processing systems, 2019, 32.

[2] Psenka, Michael, et al. Learning a diffusion model policy from rewards via Q-score matching. Proceedings of the 41st International Conference on Machine Learning. 2024.

**Questions:**

1. In the training curves in Figure 3, why does DACERv2 improve more slowly than most baselines in the early stage, especially on Ant-v3, HalfCheetah-v3, Walker2d-v3, and Swimmer-v3?
2. The argument in Lines 321-323 is not logically supported. Why does the auxiliary intermediate supervisory signal enable action sampling with fewer diffusion steps? A similar Q-gradient in QSM still requires 20 sampling steps.

If the authors can address the concerns above, I would be willing to increase the overall score.

---

> ### Author Response · Authors · 2025-11-21
> **(1/2)  Re: the Reviewer n3U9**
>
> We thank you for the careful reading of our paper and constructive comments in detail.
>
> # 1. Different score function forms and slow mixing issues (Weakness1)
> Thank you for the insightful observation regarding the score function. We would like to clarify the **fundamental distinction** between our method and standard diffusion models.
>
> You are correct that in standard generative diffusion models, the score function is trained to match the score of the "perturbed distribution." This distribution $p_t(a_t)$ is defined via a forward noising process:
>
> $$p_t(a_t) = \int q(a_t|a_0) p_0(a_0) da_0,$$
> where $q(a_t|a_0)$ is the transition kernel and $p_0(a_0)$ is the data distribution. The network $S_\theta$ is typically trained to approximate $\nabla_{a_t} \log p_t(a_t|s_t)$.
>
> In contrast, our Eq. (9) corresponds to the score of the target Boltzmann distribution $\pi_\text{soft}$ (our desired policy), which is derived directly from the Q-function: $\nabla_{a}\log~\pi(a|s)=\frac{1}{\alpha}\nabla_{a}Q(s,a)$.
>
> The core distinction is that **neither DACER nor DACERv2 constructs** a standard diffusion model based on denoising a forward process. The DACER series solely leverages the reverse-time SDE (Eq. 5) as an expressive policy approximator. Crucially, we **do not define a forward noising process** that maps data $a_0$ to noise $a_T$. Consequently, the "perturbed distribution" $p_t(a_t)$ mentioned in your comment is undefined in our framework.
>
> Therefore, $S_\theta(s, a_t, t)$ in our policy SDE **does not approximate** $\nabla_{a_t} \log p_t(a_t|s_t)$. Instead, it functions as a **noise prediction network**, optimized to guide the reverse sampling process to converge to the target policy.
>
> ---
>
> We acknowledge the reviewer's concern regarding the slow mixing of non-annealed Langevin dynamics, as noted in [1]. To clarify, DACERv2 does not employ non-annealed Langevin dynamics as its sampling policy. We discuss Langevin dynamics in Section 3.1 primarily to provide theoretical intuition. Our actual policy is a reverse-time diffusion SDE, as defined in Eq. (5). Crucially, **this SDE operates with a time-dependent noise schedule, effectively serving as an annealing mechanism.**
>
>
> # 2. Relationship between two objectives (Weakness2)
> We thank the reviewer for this critical insight. We respectfully clarify that while the combination may appear structurally simple, the two objectives are synergistic rather than conflicting. Fundamentally, both serve a unified goal: training the diffusion policy $\pi_\theta$ to approximate the optimal Boltzmann distribution, $\pi^*(a|s) \propto \exp(Q(s,a))$. Our method approaches this goal through two complementary mechanisms.
>
> The original DACER loss, $\mathcal{L}_q$, serves as the terminal objective by maximizing the Q-value of the final generated action. While this ensures the policy produces high-quality outcomes, the gradient signal must backpropagate through the entire denoising chain, often resulting in attenuation issues similar to the vanishing gradient problem in RNNs [2]. To address this, the Q-gradient loss, $\mathcal{L}_g$, provides intermediate supervision. Leveraging the theoretical insight that the optimal score function is proportional to $\nabla_a Q(s,a)$, $\mathcal{L}_g$ directly aligns the policy's denoising dynamics with the geometry of the Q-function. Consequently, rather than conflicting, $\mathcal{L}_g$ provides dense, direct gradients to shape the diffusion process—effectively mitigating the attenuation in $\mathcal{L}_q$—while $\mathcal{L}_q$ ensures the global optimization direction remains anchored to value maximization.
>
> # 3. Description (Weakness3)
> Thank you for your reminder. This statement may have been misleading, and we have revised it. As described in Section 2.1, policy multimodality is achieved by combining Q-value maximization and entropy regularization with a diffusion model.
>
> # 4. The performance improvement rate is slow in the early stages of some tasks (Question1)
> Your comment is very detailed and insightful. In the original DACER, the Q-value maximization objective only utilizes the final output action $a_0$ from the diffusion model. The additional Q-graident loss, however, **requires the Q-gradient field to provide estimations using the intermediate actions $a_t$ from the entire diffusion process as input**. Therefore, our method places a higher demand on the Q-value learning, which is reflected in the training curves as a slower rate of improvement during the early stages compared to the original DACER.

---

> ### Author Response · Authors · 2025-11-21
> **(2/2) Re: the Reviewer n3U9**
>
> # 5. The reason why Q-gradient can improve efficiency in DACERv2 (Question2)
> Thank you for this highly insightful comment. You have correctly identified a critical point: simply utilizing a Q-gradient, as done in QSM [3], is insufficient to achieve 5-step sampling efficiency.
>
> We clarify that our efficiency gains stem not merely from the use of the Q-gradient, but from specific innovations designed to overcome its inherent limitations. Our approach distinguishes itself from QSM in three fundamental aspects, which collectively resolve the inefficiency issues:
>
> 1. **Difference in Objective.** Unlike QSM, which relies solely on score-matching to align with the Q-gradient, DACERv2 adopts a multi-task framework. It retains the original DACER's end-to-end Q-value maximization ($\mathcal{L} _ {q}$) as the primary objective to ensure a stable, global baseline. The Q-gradient field ($\mathcal{L} _ {g}$) serves as an auxiliary loss, providing intermediate supervisory signals to guide this global objective. Because this intermediate signal facilitates step-wise gradient descent in the action space, it promotes action generation that approximates monotonic refinement across the Q-function landscape—a key distinction from methods relying exclusively on terminal Q-value maximization. Consequently, DACERv2 achieves superior sample efficiency.
>
> 2. **Difference in Gradient Adaptation.** As noted in the paper, the raw Q-gradient is time-independent, presenting a mismatch with the time-dependent nature of the diffusion noise prediction network. Direct usage of this gradient (as in QSM) leads to suboptimal performance. DACERv2 resolves this mismatch via a temporal weighting mechanism, $w(t)$. This mechanism modulates the influence of the Q-gradient guidance based on the diffusion step $t$, enabling strong denoising in early stages and fine-tuning in later stages. Our ablation study in Figure 4(b) explicitly demonstrates that **removing this time-weighting mechanism results in a significant performance drop**, confirming its necessity for our efficiency.
>
> 3. **Difference in Gradient Stability.** We also observed that raw Q-gradients can be volatile, potentially destabilizing training. DACERv2 stabilizes this auxiliary loss through gradient normalization. As shown in Appendix F (Figure 13), this normalization is beneficial for performance.
>
> In summary, you are correct that a naive application of the Q-gradient is inefficient. Instead, we employ it as **an auxiliary to a stronger max-Q objective and, crucially, adapt it for the diffusion process via temporal weighting and normalization.** This combination remedies the limitations of the raw Q-gradient seen in QSM, allowing us to achieve high performance with significantly fewer steps.
>
>
> # Reference
> [1] Song, Yang, and Stefano Ermon. "Generative modeling by estimating gradients of the data distribution." Advances in neural information processing systems 32 (2019).
>
> [2] Pascanu, Razvan, Tomas Mikolov, and Yoshua Bengio. "On the difficulty of training recurrent neural networks." International conference on machine learning. Pmlr, 2013.
>
> [3] Psenka, Michael, et al. "Learning a diffusion model policy from rewards via q-score matching." arXiv preprint arXiv:2312.11752 (2023).

---

> ### Author Response · Authors · 2025-11-26
>
> Thank you again for the great efforts and valuable comments. We have carefully addressed the main concerns in detail. We hope you might find the response satisfactory. As the discussion phase is about to close, we are very much looking forward to hearing from you about any further feedback. We will be very happy to clarify any further concerns (if any).

---

### Official Review · Reviewer_eLPu · 2025-11-01

**Soundness:** 2
**Presentation:** 2
**Contribution:** 2
**Rating:** 4
**Confidence:** 4

**Summary:**

The paper introduces the **DACER v2** algorithm, an improved version of **DACER**, which incorporates a **Q-gradient field** and a **temporal weighting mechanism**. Experiments on **OpenAI Gym** benchmarks and **multi-modal tasks** demonstrate that **DACER v2** achieves superior performance and stronger multi-modality using only **five diffusion denoising steps**, outperforming other online diffusion RL algorithms.

**Strengths:**

1. The paper addresses a highly important and timely research question, especially as diffusion models are becoming increasingly dominant in the fields of **imitation learning**, **reinforcement learning**, and **Vision-Language-Action (VLA)** modeling.

2. The paper is **well-written** and **easy to follow**, presenting its ideas and methodologies clearly.

3. The **DACER v2** algorithm demonstrates **strong performance** compared to other **online diffusion RL** methods.

**Weaknesses:**

### Major Weaknesses:

1. The authors claim that the **DACER v2** algorithm focuses on improving the diffusion efficiency of the original **DACER**. Accordingly, one would expect **DACER v2** to achieve comparable performance with fewer diffusion denoising steps compared to **DACER** using the full number of steps. However, the experimental results show that **DACER v2** not only maintains efficiency but also exhibits **stronger multi-modality** and **better sample efficiency** with fewer denoising steps. The authors are encouraged to explain the source of this additional performance gain in more detail.

2. The main experimental figure (**Figure 3**) presents total average return plotted against **iterations**. Could the authors clarify what these iterations represent? Are they equivalent to **environment timesteps**? If not, please explain the rationale behind using this setting and consider including additional plots showing **total average return versus environment timesteps** for clearer comparison.

3. In **Section 4.2**, the authors claim that *“in real-time industrial control tasks, the inference time should be less than 1 millisecond to meet control requirements.”* It would be helpful to provide additional evidence or references to substantiate this statement. From my perspective, an inference time of **1.6 ms** (as achieved by the original **DACER** algorithm) already appears sufficient for most robotic control tasks, and further reducing it to **0.6 ms** may offer only marginal benefits. Since this point directly relates to the **motivation of the paper**, a clearer justification would strengthen the argument.

4. There exists a wide range of **diffusion acceleration methods**, such as **DDIM** and **Consistency Models**. In the introduction, the authors claim that these acceleration techniques trade performance for efficiency. It is highly recommended that the authors evaluate **DACER** combined with a diffusion acceleration method during inference and compare the results with **DACER v2**, to more clearly demonstrate the advantages of the proposed approach.

5. The authors are encouraged to include a discussion or experimental comparison with **Diffusion Policy Policy Optimization (DPPO)**, as it represents a closely related and widely used approach in online diffusion RL.


### Minor Weaknesses:

1. The authors are recommended to discuss some highly related works:

**Diffusion Acceleration:**

Song, Jiaming, Chenlin Meng, and Stefano Ermon. "Denoising diffusion implicit models." arXiv preprint arXiv:2010.02502 (2020).

Song, Yang, et al. "Consistency models." (2023).

**Online RL with Diffusion Policy:**

Yuan, Xiu, et al. "Policy decorator: Model-agnostic online refinement for large policy model." arXiv preprint arXiv:2412.13630 (2024).

Ankile, Lars, et al. "From imitation to refinement-residual rl for precise assembly." 2025 IEEE International Conference on Robotics and Automation (ICRA). IEEE, 2025.

Wagenmaker, Andrew, et al. "Steering Your Diffusion Policy with Latent Space Reinforcement Learning." arXiv preprint arXiv:2506.15799 (2025).

**Offline RL with Diffusion Policy:**

Hansen-Estruch, Philippe, et al. "Idql: Implicit q-learning as an actor-critic method with diffusion policies." arXiv preprint arXiv:2304.10573 (2023).

Park, Seohong, Qiyang Li, and Sergey Levine. "Flow q-learning." arXiv preprint arXiv:2502.02538 (2025).

**I am more than willing to raise my scores if the authors adequately address my concerns**

**Questions:**

1. (Related to Major Weakness 1) Where do the observed improvements in **multi-modality** and **sample efficiency** originate from? Is **DACER v2** a strictly superior algorithm compared to the original **DACER** ?

2. (Related to Major Weakness 2) What do the **iterations** in **Figure 3** represent?

3. (Related to Major Weakness 3) Why do the authors believe that an inference time of **less than 1 millisecond** is required to meet **real-time industrial control** requirements?

---

> ### Author Response · Authors · 2025-11-21
> **(1/3) Re: the Reviewer eLPu**
>
> We sincerely appreciate your detailed and constructive feedback. Below are our responses to your concerns.
>
> # 1. Explanation for DACER v2’s performance gains with fewer diffusion steps (Weakness1 & Q1)
> We sincerely appreciate your insightful comment. In the original DACER, the Q-function is maximized exclusively at the final denoising step, serving as **a terminal optimization signal**. As a result, gradients back-propagated to earlier diffusion steps act similarly to RNNs [1] in long-horizon tasks, becoming **increasingly indirect and attenuated**, which limits the efficacy of policy updates. To mitigate this, we introduce **intermediate supervision** at every diffusion step. We leverage the theoretical insight that the optimal policy in DACER (derived from entropy-constrained Q-maximization) and the policy induced by Langevin dynamics share a consistent Boltzmann form parameterized by the value networks. By incorporating this supervision at each step, the policy receives **a significantly denser and more direct learning signal**. This enables DACERv2 to converge more effectively toward the value function's Boltzmann distribution, thereby achieving superior multi-modality compared to the original DACER. Moreover, as the intermediate signal ensures step-wise gradient descent in the action space, **it facilitates an action generation process that approximates monotonic refinement across the Q-function landscape, in contrast to methods using only terminal loss.** Consequently, DACERv2 achieves higher sample efficiency and performance, even with fewer denoising steps.
>
> # 2. The meaning of iteration (Weakness2 & Q2)
> For the sake of fairness in comparison, we maintained the same experimental setup and plotting standards as used in DSAC [2] and DACER. In DSAC and DACER, **each iteration represents one network update, and 20 samples are sampled per iteration**. For env steps, you need to multiply the horizontal axis by 20.
>
> # 3. Questions about efficiency statements (Weakness3 & Q3)
> We acknowledge that the statement you highlighted lacked sufficient rigor and have removed it. Our central argument is that **reducing the number of denoising steps is imperative for high-frequency control in real-world applications**, particularly given hardware constraints (e.g., on industrial computing platforms). For instance, diffusion-based planners in autonomous driving typically utilize five or fewer denoising steps to adhere to real-time latency budgets [3]. Similarly, in automotive suspension control systems employing magnetorheological damping, the Electronic Control Unit (ECU) **operates** at approximately 1000 Hz [4]. Furthermore, the computational advantage of fewer diffusion steps becomes increasingly significant **as the scale of the policy network increases**.
>
> # 4. Integrating diffusion acceleration methods into original DACER (Weakness4)
> Your suggestions can help make the advantages of our approach more obvious. Taking the HalfCheetah-v3 task as an example, we equipped original DACER with DDIM and Consistency Model methods separately, **reducing diffuision steps from 20 to 5**. The experimental results are shown in Table 1. The results show that, with a diffusion step size of 5 steps, the performance of DACERv2 is **improved by 131.1% and 45.0% compared to the original DACER with DDIM and Consistency Model, respectively**.
>
> Table 1 **Total Average Return**. The mean value for each method is bolded. ± corresponds to standard deviation over five runs.
> | Method                                | HalfCheetah-v3  |
> | ------------------------------------- | --------------- |
> | DACERv2                               | **18192 ± 266** |
> | original DACER                        | 17488 ± 216     |
> | original DACER with DDIM              | 7872 ± 344      |
> | original DACER with Consistency Model | 12542 ± 281     |
>
> Unlike conventional approaches that accelerate diffusion policies primarily by optimizing the sampling mechanism, our method **directly tackles the lack of intermediate supervision** during the training phase. By resolving this issue, we significantly enhance single-step generation quality, striking a superior balance between policy performance and computational cost.

---

> ### Author Response · Authors · 2025-11-21
> **(2/3) Re: the Reviewer eLPu**
>
> # 5. Comparison of DPPO (Weakness5)
> Thank you for the insightful suggestions. We have incorporated a discussion of DPPO into the related work sections. DPPO is recognized for being the first to successfully apply policy gradient methods to the fine-tuning of diffusion policies. Its core contribution is the introduction of a "Two-Layer MDP" optimization perspective. This framework models the policy's iterative denoising process as an inner MDP and the agent-environment interaction as the outer MDP. This formulation uniquely enables the end-to-end optimization of the diffusion policy using standard policy gradient algorithms.
>
> ## Similarities:
> In the lineage of diffusion policy development, both DPPO and DACERv2 are built upon the consensus of using the reverse diffusion process as the policy generation mechanism. The action distribution is generated via this iterative denoising process, which inherently endows the models with capabilities for structured exploration, multi-modal outputs, and strong generalization. Broadly, DPPO and the DACER series are both representative approaches advancing the integration of diffusion policies with online RL. DPPO champions the PPO-style (on-policy) approach, whereas DACERv2 exemplifies the Actor-Critic (off-policy) approach.
>
> ## Differences:
> The core contribution of DPPO lies in its "diffusion process as a two-layer MDP" optimization perspective. This framework is what **enables the seamless integration of diffusion models with PPO-style RL algorithms**. DPPO emphasizes the fine-tuning of later denoising steps, stable structured exploration, and the high-quality on-policy adaptation of demonstration-pretrained policies.
>
> In contrast, **DACERv2 adheres to an off-policy Actor-Critic framework**. It emphasizes critic learning via a Q-function, utilizes entropy regularization, and introduces a "Q-gradient field objective. The primary focus of DACERv2 is to address **the critical efficiency bottleneck of using diffusion policies in online RL**.
>
> # 6. Supplement related paper (Minor Weakness)
> Thank you for your important information. We have added these paper to the related work section. The following is the added content.
>
> ## Diffusion Acceleration
> The pursuit of efficient diffusion sampling has yielded several key advancements. Denoising Diffusion Implicit Models (DDIM)[5] re-envisioned the reverse process as a deterministic ODE, permitting significant sampling speed-ups. DPM-Solver[6] introduced high-order exponential integrators, achieving high-fidelity generation without retraining. DPM-Solver++[7] further adapted this high-order approach for the widely-used classifier-free guidance regime, stabilizing sampling at large guidance scales. Concurrently, Consistency Models[8] explored a distillation-based approach, compressing the multi-step ODE trajectory into a single ``consistency function" that maps noise to data in one or few steps.
>
>
> ## Online RL with Diffusion Policy
> DPPO[9] formulates the reverse diffusion process as a secondary MDP and applies on-policy PPO-style optimization, achieving strong performance. Yuan \textit{et al.} proposed Policy Decorator[10], which treats a large base policy (e.g., Diffusion Policy) as a black box and learns a bounded residual policy with PPO to improve performance in a model-agnostic and stable manner. Ankile \textit{et al.} introduced ResiP[11], which regards a chunked imitation policy as a high-level planner and trains a closed-loop residual controller to provide fine-grained corrections for precise assembly. Wagenmaker \textit{et al.} proposed DSRL[12], which steers a frozen diffusion policy by running RL in its latent noise space with a dual-$Q$ architecture, achieving sample-efficient online adaptation without finetuning the diffusion network weights.
>
>
> ## Offline RL with Diffusion Policy
> In parallel, Hansen-Estruch \textit{et al.} proposed IDQL[13], which reinterprets IQL as a behaviour-regularised actor-critic method and uses a diffusion-model among the behaviour cloning policy to extract the implicit actor. Recently, Park \textit{et al.} proposed Flow Q-Learning (FQL)[14], which leverages an expressive flow-matching policy together with a separately RL-trained one-step actor to model complex action distributions without backpropagating through iterative generation, achieving competitive results across large-scale offline and offline-to-online benchmarks.

---

> ### Author Response · Authors · 2025-11-21
> **(3/3) Re: the Reviewer eLPu**
>
> # Reference
> [1] Pascanu, Razvan, Tomas Mikolov, and Yoshua Bengio. "On the difficulty of training recurrent neural networks." International conference on machine learning. Pmlr, 2013.
>
> [2] Duan, Jingliang, et al. "Distributional soft actor-critic with three refinements." IEEE Transactions on Pattern Analysis and Machine Intelligence (2025).
>
> [3] Liao, Bencheng, et al. "Diffusiondrive: Truncated diffusion model for end-to-end autonomous driving." Proceedings of the Computer Vision and Pattern Recognition Conference. 2025.
>
> [4] https://www.bwigroup.com/wp-content/uploads/2025/06/b9a6f231e2fcf8666b5501f2961b228fe732cf50.2-strony-2-1-1.pdf
>
> [5] Song, Jiaming, Chenlin Meng, and Stefano Ermon. "Denoising diffusion implicit models." arXiv preprint arXiv:2010.02502 (2020).
>
> [6] Lu, Cheng, et al. "Dpm-solver: A fast ode solver for diffusion probabilistic model sampling in around 10 steps." Advances in neural information processing systems 35 (2022): 5775-5787.
>
> [7] Lu, Cheng, et al. "Dpm-solver++: Fast solver for guided sampling of diffusion probabilistic models." Machine Intelligence Research (2025): 1-22.
>
> [8] Song, Yang, et al. "Consistency models." (2023).
>
> [9] Ren, Allen Z., et al. "Diffusion policy policy optimization." arXiv preprint arXiv:2409.00588 (2024).
>
> [10] Yuan, Xiu, et al. "Policy decorator: Model-agnostic online refinement  for large policy model." arXiv preprint arXiv:2412.13630 (2024).
>
> [11] Ankile, Lars, et al. "From imitation to refinement-residual rl for  precise assembly." 2025 IEEE International Conference on Robotics and  Automation (ICRA). IEEE, 2025.
>
> [12] Wagenmaker, Andrew, et al. "Steering Your Diffusion Policy with Latent  Space Reinforcement Learning." arXiv preprint arXiv:2506.15799 (2025).
>
> [13] Hansen-Estruch, Philippe, et al. "Idql: Implicit q-learning as an actor-critic method with diffusion policies." arXiv preprint arXiv:2304.10573 (2023).
>
> [14] Park, Seohong, Qiyang Li, and Sergey Levine. "Flow q-learning." arXiv preprint arXiv:2502.02538 (2025).

---

> ### Author Response · Authors · 2025-11-26
>
> Thank you again for the great efforts and valuable comments. We have carefully addressed the main concerns in detail. We hope you might find the response satisfactory. As the discussion phase is about to close, we are very much looking forward to hearing from you about any further feedback. We will be very happy to clarify any further concerns (if any).

---

### Public Comment · ~Haque_Ishfaq1 · 2025-11-16
**Comparing against Langevin Soft Actor Critic  from ICLR 2025 and DeepMind Control Suite experiment suggestion**

Dear authors,

We wanted to point out that our ICLR 2025 paper on Langevin Soft Actor Critic [1] should be used as a baseline. In LSAC, we use Langevin Monte Carlo (LMC) based Thompson Sampling and distributional critic to achieve sample efficiency as well as multi-modal behavior through parallel tempering based LMC. Given one of the main motivations for this submission is achieving multi-modal and expressive behavior in the policy, we believe our LSAC paper is highly relevant as an alternative baseline to diffusion policy based approaches.

In our work, we used `DSAC-T` (Duan et al., 2023), `QSM` (Psenka et al., 2024), `DIPO` (Yang et al., 2023), `SAC` (Haarnoja et al., 2018a), `TD3` (Fujimoto et al., 2018), `PPO` (Schulman et al., 2017), `TRPO` (Schulman et al., 2015), `REDQ` (Chen et al., 2021),  `DrQ-v2` (Yarats et al., 2022)) and model-based (`Dreamer` (Hafner et al., 2020)) as baselines and we got superior results than these baselines in MuJoCo tasks and DeepMind Control Suite tasks. We want to **emphasize** that `LSAC` outperforms baselines `QSM`, `DIPO`, `PPO`, `SAC` and `DSAC` which were also used as baseline in this submission. Note that in MuJoCo `v-3`, which is same as what the authors here used, we used `10 seeds` and `1 million` time steps. We encourage the authors to use `10 seeds` for more robust and conclusive evidence. Our codebase and **all the data** are **publicly** available here https://github.com/hmishfaq/LSAC

### **Suggestion for further experiment in DeepMind Control Suite (DMC) (Tassa et al., 2018)**

We would also like to suggest the authors to run their algorithm in DeepMind Control Suite (DMC) (Tassa et al., 2018) environments which are standard benchmarking environments for continuous control tasks. As mentioned above, you can readily find the data for `10 seeds` over `3e6` training steps in `12` environments for `LSAC`, `DSAC`, `DIPO`, `Dreamer`, `DrQ-v2`, `TD3`, `PPO`, `SAC` and `TRPO` in our codebase https://github.com/hmishfaq/LSAC .

We hope you will find these suggestions and pointers to baseline data helpful in regard to improving your paper.

Thanks!

[1] Ishfaq, Haque, et al. "Langevin Soft Actor-Critic: Efficient Exploration through Uncertainty-Driven Critic Learning."  ICLR 2025

---

### Meta-Review · Area_Chair_dtuP · 2025-12-30

**Summary:**

The paper proposes DACERv2, an online reinforcement learning algorithm that improves the original DACER by incorporating a Q-gradient field objective and a temporal weighting mechanism. While the paper demonstrates strong empirical performance on the benchmarks, the reviewers raised concerns regarding the theoretical clarity, novelty of the work, and insufficient baselines. Though I do agree some reviews might contain factual errors and appreciate the authors feedback, I still feel the paper has certain room for improvement. Based on these concerns, I think the paper is not ready for publication yet and would recommend a rejection.

**Reviewer Concerns:**

Concerns addressed by the rebuttal
- Clarifications on the performance gain: The authors clarified that the performance gain comes from "intermediate supervision" at every diffusion step, which provides denser learning signals than the original DACER
- Additional baselines and related works: The authors integrated DDIM and Consistency Models into DACER for comparison and added a discussion of DPPO to the paper.
- New theoretical derivations: The authors provide a new derivation demonstrating that incorporating $L_g$ increases the Signal-to-Noise Ratio (SNR) of the gradients compared to using $L_q$ alone.

Concerns not addressed
- The novelty of the work, as discussed by reviewer VwAt and dk8y, is limited. From my point of view, the authors clarifications do not fully address this concern
- Sensitivity to hyper parameters. As pointed out by reviewer dk8y, the method might be quite sensitive to hyperparameters. Authors also acknowledge this limitation.
- Insufficient justifications to the gradient conflict issue. Although authors provide expalantions to reviewer n3U9's question regarding the gradient conflicting issue, it is not justified by any supporting evidences.

**Reviewer Scores:**

I think eLPu might raise the score to 5 as many of eLPu's questions have been addressed.

---

### Decision · Program_Chairs · 2026-01-26

Reject